# Generative Data Augmentation via Diffusion Distillation, Adversarial Alignment, and Importance Reweighting

**Ruyi An**[1*]    **Haicheng Huang**[2*]    **Huangjie Zheng**[3]    **Mingyuan Zhou**[1]
[1]The University of Texas at Austin    [2]Shanghai Jiao Tong University    [3]Apple
ruyian@utexas.edu  mingyuan.zhou@mccombs.utexas.edu

## Abstract

Generative data augmentation (GDA) leverages generative models to enrich training sets with entirely new samples drawn from the modeled data distribution to achieve performance gains. However, the usage of the mighty contemporary diffusion models in GDA remains impractical: *i)* their thousand-step sampling loop inflates wall-time and energy cost per image augmentation; and *ii)* the divergence between synthetic and real distributions is unknown–classifier trained on synthetic receive biased gradients. We propose DAR-GDA, a three-stage augmentation pipeline that unites model **D**istillation, **A**dversarial alignment, and importance **R**eweighting that makes diffusion-quality augmentation both fast and optimized for improving downstream learning outcomes. In particular, a teacher diffusion model is compressed into a one-step student via score distillation, slashing the time per-image cost by $> 100\times$ while preserving FID. During this distillation (D), the student model additionally undergoes adversarial alignment (A) by receiving direct training signals against real images, supplementing the teacher's guidance to better match the true data distribution. The discriminator from this adversarial process inherently learns to assess the synthetic-to-real data gap. Its calibrated probabilistic outputs are then employed in reweighting (R) by importance weights that quantify the distributional gap and adjust the empirical loss when training downstream models; we show that reweighting yields an unbiased stochastic estimator of the real-data risk, fostering training dynamics akin to those of genuine samples. Experiments validate DAR-GDA's synergistic design through progressive accuracy gains with each D-A-R stage. Our approach not only surpasses conventional non-foundation-model GDA baselines but also remarkably matches or exceeds the GDA performance of large, web-pretrained text-to-image models, despite using solely in-domain data. DAR-GDA thus offers diffusion-fidelity GDA samples efficiently, while correcting synthetic-to-real bias to benefit downstream tasks.

## 1 Introduction

From early datasets of a few hundred instances [23, 22, 63] to today's web-scale corpora [17, 57, 74], data has always been the engine of machine learning. Discriminative models are only as good as the quantity, quality, and diversity of the samples they see. Data augmentation eases this dependency by synthesizing additional training examples, aiming to improve the generalization and robustness of models by exposing them to a more varied set of training instances. Classic hand-crafted transforms, such as geometric warps, color jitter, flips, exploit invariances of natural images [51, 78] but cannot create entirely new content. Generative data augmentation (GDA) closes this gap by sampling fresh, high-fidelity images that improve downstream performance and unlock learning under privacy [112, 95, 35], security [108, 88], or copyright constraints [56, 76].

---

* These authors contributed equally.

39th Conference on Neural Information Processing Systems (NeurIPS 2025).

Among generative models, diffusion models [31, 85] have emerged as the state-of-the-art, delivering unparalleled sample quality and mode coverage [18, 32]. However, two obstacles limit their practical use for augmentation: *i)* diffusion sampling requires hundreds to thousands of denoising iterations, significantly inflating computational cost per image [105]; *ii)* the generator distribution $q_G$ inherently differs from the true data distribution $p_{\text{data}}$. Unmeasured discrepancies between $q_G$ and $p_{\text{data}}$ can seed spurious artifacts [19] and coverage bias [94, 69], and ultimately skew classifier gradients, leading to biased generalization.

To overcome these challenges and unlock the potential of diffusion models for GDA, we introduce DAR-GDA, a unified framework that renders diffusion-quality, efficient generative augmentation by intertwining diffusion model **D**istillation, **A**dversarial alignment, and **R**eweighting by importance in GDA. First, score-based **D**istillation compresses a full teacher diffusion model trained on the true dataset into an efficient, single-step student generator, drastically cutting wall-time per sample by more than two orders of magnitude while preserving the Fréchet Inception Distance. Second, when integrated with the distillation process, **A**dversarial training further aligns the student model with the underlying distribution: the student generator not only imitates the teacher's scores but also competes against real images, which directly minimizes the Jensen–Shannon divergence with the true data, further narrowing the gap between $q_G$ and $p_{\text{data}}$. Crucially, the discriminator learned in this process doubles as a density-ratio estimator; its calibrated output approximates the density ratio between two distributions, yielding per-sample importance weights. Third, for **R**eweighting, these importance weights are applied when training downstream learners on synthesized images from the fixed student generator, for which reweighting the classification objective with these importance weights yields an unbiased stochastic estimator of the true data risk. Furthermore, we show that the combined effect of adversarial alignment and importance reweighting tightens an upper bound on generalization error relative to conventional GDA. While the proposed DAR-GDA framework is generally applicable to discriminative modeling, in this work, we focus on one specific task of classification as the fundamental benchmark for investigating its impact on model generalization. The source code is available at `https://github.com/ruyianry/gda-dar`.

Our contributions are as follows:

- We introduce DAR-GDA, a unified framework enabling practical GDA with diffusion models by concurrently addressing i) their slow sampling speeds and ii) the synthetic-to-real data bias, integrated via adversarial training.
- We demonstrate how integrating adversarial training with score distillation not only improves the distilled model through real-data guidance by minimizing JS divergence, but also yields a discriminator. This discriminator, in turn, enables a sample importance reweighting to correct for bias present in empirical risk on synthetic samples, forming our three-stage DAR-GDA approach.
- Experiments confirm progressive performance gains across D-A-R stages, with DAR-GDA simultaneously demonstrating superior classification performance and efficiency for GDA on CIFAR-10 and ImageNet-1K.

## 2 Related work

### 2.1 Generative Data Augmentation

Early GDA approaches rely on VAEs and GANs [48, 25, 9], which proved valuable for niche scenarios such as class imbalance, low-shot recognition, and domain shifts [120, 91, 73, 77, 7, 67, 54, 115, 116, 53, 1, 36]. Their impact on large-scale, high-resolution tasks remained limited because those models struggled to capture complex data distributions. Diffusion models overcome that limitation: they faithfully reproduce fine detail and avoid mode collapse, enabling advances in robustness [6, 26, 10, 62], privacy preservation [65], data imbalance [3], and self- or semi-supervised learning [98, 90, 113, 8]. A growing line of work harnesses foundation models—diffusion generators pre-trained on web-scale corpora such as Stable Diffusion and GLIDE [68, 64]—and then aligns them to a downstream domain [5, 34, 99, 72, 123, 6, 29, 109]. While providing strong priors, this strategy raises several concerns such as licensing and usage constraints and limited applicability in fields with no web counterpart like scientific diffraction imaging [97]. Crucially, the high inference cost of iterative diffusion sampling remains, rendering it an economically challenging solution. These

limitations motivate a task-centric GDA pipeline that *i)* learns an expressive generator directly from the classifier's own training set; and *ii)* produces diffusion-quality samples at reduced cost.

## 2.2 Adversarial Training

GANs cast generation as a two-player game in which a discriminator learns to distinguish between real and generated samples, outputting a probability of a sample being real [25]. This adversarial process is recognized for minimizing the Jensen-Shannon divergence between the true data distribution and the generator's distribution, which is distinct from the reconstruction losses-based generative models [21]. Beyond its role in training the generator, the GAN discriminator's output can be interpreted as a likelihood-free density-ratio estimator [93, 27, 81, 92], which has spurred various techniques such as post-hoc discriminator-guided rejection sampling or weakly supervised discriminator trained on reference set to refine generator outputs, yielding fairer or higher-fidelity sample sets [11, 44, 52, 4]. Both rely on training a GAN from scratch and therefore inherit mode-collapse and stability issues [105] that limit their usefulness for large-scale augmentation. More recent work attaches a separately trained, GAN-style discriminator to a diffusion backbone to steer sampling [42]; this requires a distinct secondary training phase for the discriminator, and the valuable density-ratio information it learns is typically not propagated to inform downstream tasks. Collectively, we see a need for a more integrated adversarial mechanism that can be synergistically employed with diffusion models, and applied for enhancing downstream applications.

# 3 Preliminaries

## 3.1 Diffusion Models

Diffusion (or score-based) generative models [80, 31, 85, 84] construct a latent Markov chain that gradually corrupts a data point $x_0 \in \mathbb{R}^d$ into noise, and learns to reverse this process to generate new data. In the forward diffusion, a deterministic schedule $\{\alpha_t, \sigma_t\}_{t=1}^T$ mixes signal and Gaussian noise:

$$x_t = \alpha_t x_0 + \sigma_t \epsilon_t, \qquad \epsilon_t \sim \mathcal{N}(\mathbf{0}, \mathbf{I}), \tag{1}$$

with $\alpha_t$ decreases and $\sigma_t$ increases with $t$ such that $x_T$ approaches standard Gaussian noise.

The reverse process is modeled by a parameterized conditional distribution $p_\phi(x_{t-1}|x_t)$ whose noise-prediction network $\epsilon_\phi$ is trained to predict the noise $\epsilon_t$ added at $t$ [31, 85]:

$$\mathcal{L}^{(\text{diffusion})}(\phi) = \mathbb{E}_{x_0 \sim \mathcal{P}, \epsilon_t, t}\big[\|\epsilon_t - \epsilon_\phi(x_t, t)\|_2^2\big], \tag{2}$$

The trained noise predictor $\epsilon_\phi$ is directly related to the score function $\nabla_{x_t} \log p_\phi(x_t)$, often approximated as $s_\phi(x_t, t) = -\sigma_t^{-1} \epsilon_\phi(x_t, t)$. To generate a new sample, the model iteratively applies this reverse process, starting from pure noise $x_T$ and gradually denoising it using the learned score over $T$ steps to reconstruct an initial data point $x_0$. While this iterative process yields high-fidelity samples, performing $T$ (often hundreds to thousands) steps incurs considerable computational cost.

## 3.2 Score Distillation

Score distillation aims to accelerate the inference of the trained $T$-step teacher diffusion sampler by condensing them into an efficient student generator, $G_\theta$. This student, often capable of few- or single-step synthesis, is trained to replicate the teacher's learned data distribution, typically by using the teacher's score predictions to guide the student's outputs at various noise levels. This is commonly obtained by training an auxiliary fake diffusion model, parameterized by $\psi$, on student output,

$$y_t = \alpha_t G_\theta(z) + \sigma_t \epsilon_t, \qquad z, \epsilon_t \sim \mathcal{N}(\mathbf{0}, \mathbf{I}) \tag{3}$$

via the same objective as Eq. 2 [59, 111, 122, 106], *i.e.*, $\mathcal{L}^{(\text{fake score})}(\psi) = \mathbb{E}_{z, \epsilon_t, t}\big[\|\epsilon_t - \epsilon_\psi(y_t, t)\|_2^2\big]$. Let $\psi^*(\theta)$ be the minimizer w.r.t. $\psi$ for fixed $\theta$. The generator is then updated by minimizing the score distill (SD) loss, the divergence $\mathcal{D}$ between the fake score $p_{\psi^*(\theta)}$ and the teacher score $p_\phi$:

$$\mathcal{L}^{(\text{SD})}(\theta) = \mathbb{E}_{z, \epsilon_t, t}\big[\mathcal{D}\big(p_{\psi^*(\theta)}(y_t; t) \,\|\, p_\phi(y_t; t)\big)\big]. \tag{4}$$

The divergence $\mathcal{D}$ can be instantiated as a Kullback-Leibler (KL) divergence [101, 59, 111], Fisher divergence [122, 121], or score-constructed trajectory-level divergence [43]. In practice, $\psi$ and $\theta$ are initialized from $\phi$ and alternately updated. This process can yield a one-step generator reproducing the teacher's quality at significantly reduced sampling cost [59, 122, 121], offering a promising path to overcome the efficiency bottleneck of diffusion models in practical GDA.

## 3.3 Generalization Risk in Supervised Learning

Let $\mathcal{P}$ be the unknown underlying data distribution over input-label pairs $(\boldsymbol{x}, y) \in \mathcal{X} \times \mathcal{Y}$. Given a hypothesis $h : \mathcal{X} \to \mathbb{R}^{|\mathcal{Y}|} \in \mathcal{H}$, *e.g.*, a $\omega$-parameterized neural network function $h_\omega$, the generalization risk $\mathcal{R}(h; \mathcal{P})$ quantifies the expected loss on unseen data using a loss function $\ell : \mathcal{Z} \times \mathcal{Y} \to \mathbb{R}$(such as cross-entropy), quantifying the penalty for the predicted output $h(x)$ given the true label $y$.

Since $\mathcal{P}$ is inaccessible, learning relies on a finite i.i.d. sample set $\mathcal{S}_\mathcal{P} = \{(x_i, y_i)\}_{i=1}^n \sim \mathcal{P}$. The true risk is then approximated by the empirical risk, an unbiased estimation $\mathcal{R}_{\text{emp}}(h; \mathcal{S}_\mathcal{P})$:

$$\mathcal{R}(h; \mathcal{P}) = \mathbb{E}_\mathcal{P}\big[\ell\big(h(x), y\big)\big] \approx n^{-1} \sum_{i=1}^n \ell\big(h(x_i), y_i\big) = \mathcal{R}_{\text{emp}}(h; \mathcal{S}_\mathcal{P}). \tag{5}$$

Empirical risk minimization (ERM) [96, 60] is a fundamental principle of statistical learning that aims to find a hypothesis that minimizes this empirical risk, $\mathcal{R}_{\text{emp}}$, with the expectation that a small empirical risk translates into low true risk and thus good generalization. The ERM principle forms the operational basis for the vast majority of supervised learning algorithms [61].

# 4 Distillation, Adversarial Alignment, and Reweighting for GDA

Before presenting our unified framework in Section 4.4, we first motivate its design by detailing the logical progression that leads to its components. This section begins with a critical reflection on the implicit biases in conventional GDA (Section 4.1). We then establish a principled, bias-correction mechanism via density-ratio reweighting (Section 4.2). This solution's efficacy demands a generator with high fidelity and mode coverage, pointing directly to state-of-the-art diffusion models. However, their practical use is barred by prohibitive sampling cost and the lack of a built-in density-ratio estimator. We then employ adversarial score distillation (Section 4.3) as the technique that simultaneously resolves both challenges, providing the foundation for the DAR-GDA pipeline.

## 4.1 Rethinking Learning from Generated Data Augmentation

Given the unknown data distribution $(\boldsymbol{x}, y)$, the generator $G$ defines a distribution $\mathcal{Q}_G$, from which we can draw sets of augmenting synthetic samples $\mathcal{S}_{\mathcal{Q}_G} = \{(\boldsymbol{x}_i, y_i)\} \sim \mathcal{Q}_G$. A common practice to train a hypothesis, *e.g.*, a classifier, is to minimize the empirical risk on these synthetic samples:

$$|\mathcal{S}_{\mathcal{Q}_G}|^{-1} \sum_{(\boldsymbol{x}_i, y_i) \in \mathcal{S}_{\mathcal{Q}_G}} \ell\big(h(\boldsymbol{x}_i), y_i\big) \approx \mathbb{E}_{\mathcal{Q}_G}\big[\ell\big(h(\boldsymbol{x}), y\big)\big], \tag{6}$$

This strategy implicitly assumes that the synthetic distribution $\mathcal{Q}_G$ is a faithful proxy for the true data distribution $\mathcal{P}$. However, in practice, $\mathcal{Q}_G$ inevitably differs from $\mathcal{P}$. Consequently, optimizing the objective in Eq. 6 amounts to optimizing with respect to a biased approximation of the true risk $\mathcal{R}(h; \mathcal{P}) = \mathbb{E}_{(\boldsymbol{x}, y) \sim \mathcal{P}}\big[\ell(h(\boldsymbol{x}), y)\big]$. This inherent bias reflects the error in the learning objective that arises directly from the distributional misalignment between $\mathcal{Q}_G$ and $\mathcal{P}$:

$$\Delta_{\text{bias}} = \mathbb{E}_{\mathcal{Q}_G}\big[\ell\big(h(\boldsymbol{x}), y\big)\big] - \mathbb{E}_\mathcal{P}\big[\ell\big(h(\boldsymbol{x}), y\big)\big]. \tag{7}$$

This distributional misalignment can manifest in several ways. For instance, if certain features or modes are overrepresented in $\mathcal{Q}_G$ relative to $\mathcal{P}$–that is, in regions where $q_G(\boldsymbol{x}) > p_{\text{data}}(\boldsymbol{x})$, the hypothesis $h$ may overfit to these dominant synthetic patterns, thereby neglecting rarer but potentially crucial characteristics of the true data. This issue of imbalance is a recognized challenge, even in today's advanced diffusion models [94, 69, 105]. Furthermore, the deviation is exacerbated if $\mathcal{Q}_G$ generates spurious, incoherent, or low-quality samples, *i.e.*, samples $\boldsymbol{x}_s$ for which $q_G(\boldsymbol{x}_s, y) > 0$ but $p_{\text{data}}(\boldsymbol{x}_s, y) \approx 0$. Training on such artifacts, also reported in recent diffusion models [19, 37, 45], can lead the hypothesis to learn incorrect correlations, thereby undermining the effectiveness of GDA.

Given these inherent distributional discrepancies, naively treating all synthetic samples as perfect and equally valuable representatives of $\mathcal{P}$–as implied by Eq. 6–can misguide the learning process. Therefore, to obtain a more faithful estimate of the true risk $\mathcal{R}(h; \mathcal{P})$, and thus mitigate $\Delta_{\text{bias}}(h)$, it is crucial to account for variations among individual synthetic samples in their fidelity and alignment with $\mathcal{P}$. A principled approach involves reweight the contribution of each synthetic sample in the empirical risk according to its estimated alignment. This naturally gives rise to the need for a mechanism that can quantify the degree of alignment between each synthetic sample and the true data distribution, enabling the corresponding correction of the loss terms $\ell(h(\boldsymbol{x}), y)$.

## 4.2 Density-Ratio Reweighting

A natural way to perform this quantification is to compare the marginal densities of synthetic samples under the two distributions. However, since $p_{\text{data}}(\cdot)$ is only known through its empirical distribution and $q_G(\cdot)$ is defined implicitly by the generative process, directly quantifying their discrepancy–and thus correcting for the induced learning bias–is challenging.

To circumvent the need for explicit likelihood estimation, we draw inspiration from the Generative Adversarial Network (GAN) framework [25], which provides a likelihood-free mechanism for distinguishing between real and synthetic samples. In GANs, a discriminator $D$ and a generator $G$ engage in a minimax game: $\min_G \max_D \mathbb{E}_{\boldsymbol{x} \sim p_{\text{data}}(\boldsymbol{x})}[\log D(\boldsymbol{x})] + \mathbb{E}_{\boldsymbol{x} \sim q_G(\boldsymbol{x})}[\log(1 - D(\boldsymbol{x}))]$. For a fixed generator $G$, the optimal discriminator $D^*$ satisfies [25, 114]:

$$D^*(\boldsymbol{x}) = \frac{p_{\text{data}}(\boldsymbol{x})}{p_{\text{data}}(\boldsymbol{x}) + q_G(\boldsymbol{x})}. \tag{8}$$

This optimal discriminator $D^*$ captures the probability that a sample $\boldsymbol{x}$ originates from $p_{\text{data}}(\boldsymbol{x})$ rather than $q_G(\boldsymbol{x})$, thereby providing an implicit comparison between the two densities.

Leveraging this, the density ratio $r(\boldsymbol{x})$, which compares the likelihoods of a sample $\boldsymbol{x}$ under $p_{\text{data}}$ versus $q_G$ can be derived from $D^*(\boldsymbol{x})$:

$$r(\boldsymbol{x}) := \frac{D^*(\boldsymbol{x})}{1 - D^*(\boldsymbol{x})} = \frac{p_{\text{data}}(\boldsymbol{x})/(p_{\text{data}}(\boldsymbol{x}) + q_G(\boldsymbol{x}))}{1 - p_{\text{data}}(\boldsymbol{x})/(p_{\text{data}}(\boldsymbol{x}) + q_G(\boldsymbol{x}))} = \frac{p_{\text{data}}(\boldsymbol{x})}{q_G(\boldsymbol{x})}. \tag{9}$$

This ratio acts as an importance weight briding the expectations of the two involved distributions: for any integrable $f$:

$$\mathbb{E}_{(\boldsymbol{x},y) \sim \mathcal{P}}[f(\boldsymbol{x}, y)] = \mathbb{E}_{(\boldsymbol{x},y) \sim \mathcal{Q}_G}[r(\boldsymbol{x}) f(\boldsymbol{x}, y)], \tag{10}$$

so weighting synthetic samples by $r(\boldsymbol{x})$ converts expectations under $q_G$ into those under $p_{\text{data}}$. Intuitively, an $r(\boldsymbol{x}) > 1$ amplifies regions underrepresented by $q_G$, whereas an ratio $r(\boldsymbol{x}) < 1$ downweights overrepresented or atypical synthetic examples–thereby correcting distributional misalignment.

To apply this approach to GDA, we estimate $r(\boldsymbol{x})$ via the discriminator $D$ (denote the estimate $r_D(\boldsymbol{x})$) and reweight the synthetic loss contributions:

$$\mathcal{L}^{(\text{reweight})}(h; G, D) := |\mathcal{S}_{\mathcal{Q}_G}|^{-1} \sum_{\boldsymbol{x} \in \mathcal{S}_{\mathcal{Q}_G}} r_D(\boldsymbol{x}) \ell(h(\boldsymbol{x}), y)$$

$$\approx \mathbb{E}_{\mathcal{Q}_G}[r_D(\boldsymbol{x}) \ell(h(\boldsymbol{x}), y)] = \mathbb{E}_{(\boldsymbol{x},y) \sim \mathcal{P}}[\ell(h(\boldsymbol{x}), y)] \tag{11}$$

Optimizing $h$ on synthetic samples under this reweighted objective aligns training on the synthetic distribution $\mathcal{Q}_G$ with learning directly from the true data distribution $\mathcal{P}$, thereby mitigating the bias $\Delta_{\text{bias}}(h)$ that arises from their distributional discrepancy.

In practice, $r(\boldsymbol{x})$ is estimated using a parameterized discriminator $D_\eta(\boldsymbol{x}) \approx D^*(\boldsymbol{x})$ which introduces additional variance into the loss computation. To manage the inherent bias-variance trade-off, we employ two variance reduction techniques:

*Truncation:* To prevent large $r(\boldsymbol{x})$ values from dominating the loss and destabilizing training, we clip the importance weights as $\bar{r}(\boldsymbol{x}) = \min(r(\boldsymbol{x}), \gamma)$, for a threshold $\gamma \geq 1$.

*Self-normalization:* Within each mini-batch of $k$ synthetic samples $\{\boldsymbol{x}_j\}_{j=1}^k$, we apply batch-wise self-normalization to the truncated weight: $\tilde{r}(\boldsymbol{x}) = \bar{r}(\boldsymbol{x})/\sum_{j=1}^k \bar{r}(\boldsymbol{x}_j)$. This normalizes the sum of weights in a mini-batch, further stabilizing the updates.

The efficacy of importance reweighting hinges on a well-behaved density ratio $r(\boldsymbol{x})$, which requires significant overlap between the generative ($\mathcal{Q}_G$) and true ($\mathcal{P}$) distributions [24]. Generators trained with purely adversarial objectives often fail this prerequisite, suffering from mode collapse or support mismatch, especially in the early, noise-producing training stages [117, 71, 86, 87, 38, 2, 49]. This makes their discriminators poor estimators for $r(\boldsymbol{x})$. Diffusion models, on the other hand, with their high sample quality and excellent mode coverage [83, 47], emerge as ideal candidates for $q_G$. However, they present two fundamental obstacles: *i) Efficiency:* they are computationally prohibitive for augmentation sampling; and *ii) Mechanism:* they are trained without an adversarial component, leaving no mechanism to estimate the $r(\boldsymbol{x})$ required for importance reweighting. These challenges motivate a unified approach that simultaneously addresses the efficiency problem and the density-ratio-estimation problem. We therefore bridge this by integrating adversarial training directly within

the score distillation framework to compress a pre-trained diffusion teacher model into a fast student generator, while concurrently learning the discriminator needed for the bias-correcting reweighting.

## 4.3 Adversarial Score Distillation for Diffusion Models

The Reweighting stage of our DAR-GDA framework necessitates a discriminator capable of estimating the density ratio $r(\boldsymbol{x})$. As established, vanilla diffusion models not only lack this adversarial component but are also prohibitively slow due to their iterative sampling process, leading to high per-sample costs. To address both the need for a density-ratio importance estimator and the demand for computationally efficient augmentation, we therefore bridge the score diffusion distillation procedure, detailed in Section 3.2, with adversarial training, drawing inspiration from methodologies that integrate GAN-like objectives with diffusion processes [100].

Critically, the student generator $G_\theta$ is initialized from and guided by the high-fidelity, pre-trained teacher. This provides a powerful starting point, allowing the adversarial training to bypass the unstable, error-prone early stages–where a generator starting from noise struggles to produce informative samples–that typically plague purely adversarially trained models [86, 87].

In this process, the trained iterative teacher parameterized by $\phi$ is compressed into a one-step, fast student generator $G_\theta$, while a discriminator $D_\eta$ is learned simultaneously. This is framed as the following minimax objective:

$$\min_\theta \ \max_\eta \ \lambda_1 \mathcal{L}^{(\text{SD})}_{\phi,\psi}(\theta) \ + \ \lambda_2 \mathcal{V}^{(\text{adv.})}(\theta, \eta), \tag{12}$$

where $\mathcal{L}^{(\text{SD})}_{\phi,\psi}$ is the score-distillation loss defined in Eq. 4, compelling the student $G_\theta$ to match the teacher's score estimates. The adversarial component is defined by the value function $\mathcal{V}^{(\text{adv.})}(\theta, \eta)$:

$$\mathcal{V}^{(\text{adv.})}(\theta, \eta) = \mathbb{E}_{\boldsymbol{x} \sim \mathcal{P}, \ \boldsymbol{\epsilon} \sim \mathcal{N}(\mathbf{0}, \mathbf{I})} \big[ \log D_\eta(\alpha_t \boldsymbol{x} + \sigma_t \boldsymbol{\epsilon}; t) \big] + \mathbb{E}_{\boldsymbol{z} \sim \mathcal{N}(\mathbf{0}, \mathbf{I}), \ \boldsymbol{\epsilon} \sim \mathcal{N}(\mathbf{0}, \mathbf{I})} \big[ \log \big( 1 - D_\eta(\boldsymbol{y}_t; t) \big) \big], \tag{13}$$

where $\boldsymbol{y}_t = \alpha_t G_\theta(\boldsymbol{z}) + \sigma_t \boldsymbol{\epsilon}$ is a noisy sample from the student generator $G_\theta$ at $t$, and $t \sim \pi$ is a sampling distribution over timesteps. This design allows the discriminator to compare real and fake samples across various noise levels–a design shown to enhance training stability [100, 107]. In the specific case where the adversarial game operates only on the clean image space, $\pi$ simplifies to a Dirac delta distribution centered at $t = 0$. The adversarial gradient of the student is obtained by minimizing the non-saturating negative of the second expectation in Eq. 13, *i.e.*, $\mathcal{L}^{(\text{adv.})}_G(\theta) = -\mathbb{E}_{\boldsymbol{z},t} \big[ \log D_\eta(\boldsymbol{y}_t; t) \big]$.

The discriminator $D_\eta$ can be realized as a standalone network that processes the entire input to produce a single global probability [43], or as an integrated component sharing the student's score U-Net encoder [119]. In this latter, integrated approach, a final probability is obtained by applying patch-wise aggregation to the encoder's output logits. The parameter vector $\eta$ denotes all the weights of this discriminator, including any parameters shared with the student generator.

The adversarial score distillation strategy offers several synergistic advantages. The student generator $G_\theta$ produces samples in a single step, drastically reducing computational costs compared to its iterative teacher and making diffusion models practical for GDA. The synergy of adversarial training and score distillation also drives the student distribution $q_{G_\theta}$ closer to $p_{\text{data}}$ by directly minimizing the Jensen–Shannon divergence, while preserving the teacher's high-fidelity details. Empirically, adversarial distillation has shown stable behaviour, with one-step students generally matching–and frequently reported to surpass–their teachers' quality in FID [121, 43, 110, 59]. Crucially, this process yields the discriminator $D_\eta$ as a concurrent byproduct, eliminating the need for a separate training phase. This discriminator provides the exact mechanism required for the reweighting stage, furnishing the density-ratio estimate in Eq. 9 needed to debias the downstream classifier's training.

## 4.4 The DAR-GDA Framework

Having introduced the modular components of Distillation, Adversarial alignment, and Reweighting, we now synergistically combine them into the DAR-GDA framework. The DAR-GDA framework provides a unified pipeline: an efficient student generator $G_\theta$ (from Distillation) that is aligned with $p_{\text{data}}$ (via Adversarial training), and a co-trained discriminator $D_\eta$ that enables bias-correcting reweighting. This integrated design enhances generalization by mitigating the key error sources inherent in learning from synthetic data. Furthermore, initializing $G_\theta$ from a proficient diffusion teacher significantly stabilizes adversarial training and mitigates classic GAN pathologies such

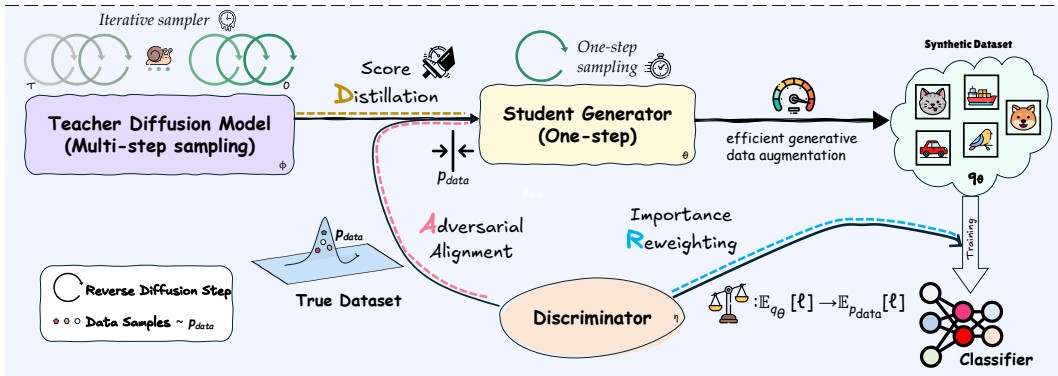

Figure 1: Overview of the DAR-GDA framework. **(D)** A multi-step teacher diffusion model is distilled into an efficient one-step student generator. **(A)** The student is adversarially aligned with the true data distribution using a discriminator. **(R)** The discriminator's outputs are then used as importance weights to train a downstream classifier.

as mode collapse and catastrophic forgetting [89]. We illustrate the overall idea in Figure 1 and summarize the complete algorithmic procedure in Appendix B.

This integrated design enhances generalization by mitigating the key error sources inherent in learning from synthetic data. We can formalize this improvement by analyzing the expected risk $\mathbb{E}_{\mathcal{P}}[\ell(h)]$ following [10, 118, 75]. This true risk can be decomposed as two summands:

$$\mathbb{E}_{\mathcal{P}}[\ell(h)] = [\mathbb{E}_{\mathcal{P}}[\ell(h)] - \mathbb{E}_{\mathcal{P}}[\ell(h)_{\mathcal{Q}_{G_\theta}}]] + [\mathbb{E}_{\mathcal{P}}[\ell(h)_{\mathcal{Q}_{G_\theta}}]], \quad (14)$$

The first summand reflects the impact of distributional mismatch. For a bounded loss $|\ell| \le L$:

$$\left|\mathbb{E}_{\mathcal{P}}[\ell(h)] - \mathbb{E}_{\mathcal{P}}[\ell(h)_{\mathcal{Q}_{G_\theta}}]\right| \le 2L\,\mathcal{D}_{\mathrm{TV}}(\mathcal{P} \parallel \mathcal{Q}_{G_\theta}) \le 2L\sqrt{2\,\mathcal{D}_{\mathrm{JS}}(\mathcal{P} \parallel \mathcal{Q}_{G_\theta})}, \quad (15)$$

by variational characterization of total variation (TV) distance and Pinsker's inequality. Adversarial score distillation directly minimizes the Jensen-Shannon (JS) divergence between $\mathcal{P}$ and $\mathcal{Q}_G$, hence tightening the upper bound on the distribution mismatch compared to using non-adversarially trained generators or the original teacher diffusion model. The second term, the biased synthetic risk, is the objective naively optimized by conventional GDA (Eq. 6). Our Reweighting stage directly corrects this. By employing the objective in Eq. 11, we replace the minimization of this biased term with an unbiased stochastic estimator of the true data risk, $\mathbb{E}_{\mathcal{P}}[\ell(h)]$.

Thus, DAR-GDA enhances generalization through a dual mechanism: the Adversarial component minimizes the distributional bias (Term 1), while the Reweighting component provides an unbiased risk estimator for hypothesis training (correcting Term 2). Coupled with the practical benefit of $> 100\times$ reduction in per-sample generation cost from Distillation, DAR-GDA is positioned as a powerful and practical tool for GDA.

## 5 Experiments

We evaluate DAR-GDA on CIFAR-10 and ImageNet-1K, comparing it with *i)* standard data-augmentation pipelines, *ii)* strong diffusion-based GDA baselines, and *iii)* a state-of-the-art GAN. The analysis proceeds in two stages. We benchmark it against conventional data augmentation methods and existing GDA baselines. We also assess its performance and adaptability with various underlying generative models. We additionally probe how DAR scales under dynamic augmentation on smaller datasets for its practical feasibility.

**Datasets.** We use the canonical train/val splits of CIFAR-10 [50] and ImageNet-1K [17]; no external or test data are introduced at any stage, including teacher training, distillation, or classification.

**Evaluation.** Classification performance is reported as top-1 accuracy for CIFAR–10 and top-1/top-5 accuracy for ImageNet-1K. Generator quality is measured with Fréchet Inception Distance (FID)[30].

**Generative Models.** For diffusion-based GDA, we adopt the publicly released checkpoints of DDPM++–EDM for CIFAR-10 [39] and DDPM++–EDM2-XXL for ImageNet-1K [40]. On CIFAR-10 we include R3GAN [33], GAN state-of-the-art achieving the best reported FID. We omit GANs on ImageNet because no open-source $512 \times 512$ model matches diffusion quality, and de novo training is

Table 1: CIFAR-10 classification accuracy with ResNet-18 and VGG-16, using training sets augmented by synthetic data equal to 1× the original dataset size (additive augmentation) on various generative models and adversarial-distillation methods under static versus dynamic data generation schemes. ∗ indicates the use of a web-pretrained text-to-image model, such as GLIDE [64] and Stable Diffusion (SD) [68] which are to be carefully compared to other generative models trained only on the original data. The best result is shown in **bold**.

| Generative Model | DAR progression | | | static (gen. once) | | dynamic (re-gen. each epoch) | |
|---|---|---|---|---|---|---|---|
| | Distill | Adv. align | Reweight | ResNet-18 | VGG-16 | ResNet-18 | VGG-16 |
| Real only | - | - | - | 95.00±0.15 | 93.76±0.20 | 95.00±0.45 | 93.76±0.50 |
| R3GAN [33] (FID=1.96) | - | ✔ | ✗ | 95.18±0.50 | 93.90±0.42 | 95.61±0.98 | 94.11±1.12 |
| | - | ✔ | ✔ | 95.32±0.47 | 93.86±0.39 | 95.72±1.00 | 93.73±1.26 |
| GLIDE + real guidance [29]* | - | - | - | 95.77±0.25 | 94.50±0.41 | - | - |
| SD + ActGen [34]* | - | - | - | 95.92±0.42 | 94.68±0.39 | - | - |
| Diffusion Model [39] (FID=1.81) | ✗ | ✗ | ✗ | 95.37±0.23 | 93.95±0.32 | 96.16±0.76 | 94.98±0.79 |
| | CTM [43] (FID=1.73) | ✗ | ✗ | 95.32±0.39 | 93.97±0.29 | 95.84±0.86 | 95.08±1.02 |
| | | ✔ | ✗ | 95.43±0.21 | 94.09±0.26 | 96.35±0.98 | 95.15±1.13 |
| | | ✔ | ✔ | 95.88±0.22 | 94.28±0.24 | 96.57±1.01 | 95.33±1.07 |
| | SiDA [121] (FID=1.39) | ✗ | ✗ | 95.48±0.25 | 94.18±0.28 | 96.29±0.68 | 95.22±0.79 |
| | | ✔ | ✗ | 95.84±0.36 | 94.53±0.36 | 96.40±1.16 | 95.40±1.02 |
| | | ✔ | ✔ | **96.21**±0.25 | **94.64**±0.20 | **96.73**±1.15 | **95.73**±1.08 |

notoriously unstable at that scale. We also include selected finetuned text-to-image models pre-trained on web-scale image data. Please note that comparisons with these models may be subject to concerns like additional information from pre-training data and potential leakage.

**Implementation Details.**

We instantiate the (D) and (A) steps of our DAR-GDA with two recent leading algorithms: CTM [43], a trajectory-based score distillation with a standalone discriminator, and SiDA [121], a Fisher-divergence-minimizing score-based distillation with encoder-sharing discriminator operating in noise space. Both are tested on CIFAR-10; SiDA alone is used on ImageNet-1K owing to CTM's current memory footprint. On CIFAR-10, we set $\alpha = 1.2$ for SiDA and train with Adam (lr=1e-5) [46] optimizer. CTM is trained for 256 steps per batch with a student learning rate of 3e-4 and the discriminator learning rate 2e-3 (batch size 128). For ImageNet-1K we distill EDM2-XXL with SiDA across 8 A100-80GB GPUs: $\alpha = 1.0$, per-GPU batch size 64, gradient accumulation every 128

Table 2: ImageNet-1K classification accuracy with ResNet-50 and ViT-S/16, using training sets augmented by synthetic data equal to 1× the original dataset size (additive augmentation) on various generative models under the static generation. ∗ denotes the use of a web-pretrained text-to-image model, Imagen [70] or Stable Diffusion (SD) [68] which are to be carefully compared to other generative models trained only on the original data. The best result is shown in **bold**.

| Generative Model | DAR progression | | | Top-1 Acc. | Top-5 Acc. |
|---|---|---|---|---|---|
| | Distill | Adv. align | Reweight | | |
| *ResNet-50 classifier* | | | | | |
| Real only | - | - | - | 76.37±0.03 | 92.86±0.08 |
| Imagen + finetune [5]* | - | - | - | 78.17 | - |
| SD + ActGen [34]* | - | - | - | **78.34**±0.32 | 94.12±0.38 |
| Diffusion Model [40] (FID=1.91) | ✗ | ✗ | ✗ | 77.12±0.15 | 93.95±0.20 |
| | SIDA [121] (FID=1.37) | ✗ | ✗ | 77.15±0.27 | 93.74±0.19 |
| | | ✔ | ✗ | 77.89±0.18 | 93.90±0.24 |
| | | ✔ | ✔ | 78.03±0.23 | **94.13**±0.32 |
| *ViT-S/16 classifier* | | | | | |
| Real only | - | - | - | 79.91±0.04 | 94.48±0.08 |
| Imagen + finetune[5]* | - | - | - | 81.00 | - |
| SD + ActGen [34]* | - | - | - | 81.17±0.51 | 95.32±0.31 |
| Diffusion Model [40] (FID=1.91) | ✗ | ✗ | ✗ | 80.50±0.18 | 95.18±0.25 |
| | SIDA [121] (FID=1.37) | ✗ | ✗ | 80.46±0.23 | 95.19±0.23 |
| | | ✔ | ✗ | 80.99±0.32 | **95.38**±0.43 |
| | | ✔ | ✔ | **81.17**±0.45 | 95.36±0.40 |

iterations using Adam (lr=5e-5) optimizer. For CIFAR-10 we train ResNet-18 [28] and VGG-16 [79]; for ImageNet-1K we use ResNet-50 [28] and ViT-S/16 [20] to evaluate the performance with and without the (R)eweighting component with self-normalization and $\gamma = 1$. CIFAR-10 models are trained for 300 epochs with batch size 128 using momentum SGD (lr=0.1). The hyperparameters $\lambda_1$ and $\lambda_2$ for the adversarial alignment objective follow the settings from prior work. On ImageNet-1K, ResNet-50 is trained for 90 epochs with batch size 4096 and initial learning rate 1.6, while ViT-S/16 is trained for 300 epochs with batch size 1024, AdamW [58], and initial learning rate 3e-3. We apply self-normalization and set $\gamma$ to be 1 for obtaining $r(x)$ for reweighting. Diffusion outputs are generated at $32 \times 32$ for CIFAR-10 and $512 \times 512$ for ImageNet; the latter are down-sampled to $224 \times 224$ before classification to match baseline protocols. Experiments use NVIDIA A100-80GB GPUs-single-GPU for CIFAR-10, 8-GPU (DDP) for ImageNet-1K implemented in PyTorch 2.1 [66].

Table 4: CIFAR-10 classification accuracy with ResNet-18 and VGG-16, using training sets *substituted by* synthetic data equal to 1× the original dataset size on various generative models and adversarial-distillation methods under static and dynamic data generation. The best result is in **bold**.

| Generative Model | DAR progression | | | static (gen. once) | | dynamic (re-gen. each epoch) | |
|---|---|---|---|---|---|---|---|
| | Distill | Adv. align | Reweight | ResNet-18 | VGG-16 | ResNet-18 | VGG-16 |
| Real only | - | - | - | 95.00±0.15 | 93.76±0.20 | 95.00±0.45 | 93.76±0.50 |
| R3GAN [33] (FID=1.96) | - | ✔ | ✗ | 90.09±0.65 | 87.75±0.39 | 90.26±0.84 | 89.79±0.58 |
| | - | ✔ | ✔ | 90.29±0.71 | 87.97±0.58 | 92.91±0.72 | 88.03±0.61 |
| Diffusion Model [39] (FID=1.81) | ✗ | ✗ | ✗ | 91.78±0.25 | 90.60±0.22 | 95.88±0.19 | 94.26±0.21 |
| CTM [43] FID=1.73 | ✗ | ✗ | ✗ | 90.83±0.39 | 89.65±0.30 | 93.70±0.24 | 93.18±0.28 |
| | ✔ | ✔ | ✗ | 91.03±0.44 | 89.78±0.46 | 94.00±0.70 | 93.27±0.67 |
| | ✔ | ✔ | ✔ | 91.37±0.43 | 90.26±0.52 | 94.32±0.72 | 93.38±0.52 |
| SiDA [121] FID=1.39 | ✗ | ✗ | ✗ | 93.20±0.28 | 91.71±0.33 | 95.78±0.35 | 94.46±0.27 |
| | ✔ | ✔ | ✗ | 93.50±0.35 | 91.88±0.38 | 96.39±0.60 | 95.25±0.52 |
| | ✔ | ✔ | ✔ | **93.77**±0.37 | **92.31**±0.48 | **96.68**±0.63 | **95.72**±0.48 |

More detailed experimental setups are provided in Appendix C. DAR-GDA thus offers a practical, drop-in GDA solution, achieving diffusion-level fidelity with efficiency bias correction.

## 5.1 Additive Augmentation Results

We evaluate DAR-GDA by supplementing the original training set with an equal volume of synthetic samples that replicate the original label distribution, using both static (one-time) and dynamic (per-epoch) generation strategies. Tables 1 and 2 present classification results for CIFAR-10 and ImageNet-1K. DAR-GDA consistently boosts accuracy across both datasets and hypothesis sets. On CIFAR-10, it improves ResNet-18 by +1.7 pp and VGG-16 by +1.9 pp. Similar gains are observed on ImageNet-1K for ResNet-50 (+1.7 pp) and ViT-S/16 (+1.3 pp). Notably, our in-domain DAR-GDA (trained solely on task-specific data) matches or outperforms pre-trained large text-to-image models, particularly where baseline diffusion models without DAR are suboptimal. Dynamic generation consistently yields better results than static, underscoring the value of increased diversity. Also, SiDA leads to better classification performance than CTM, correlating with SiDA's superior FID score.

Table 3 highlights the substantial GPU time savings for synthetic data generation. Distillation achieves synthesizing speeds comparable to fast GANs, positioning (D)istillation as a key to achieve an economical solution for high-quality GDA.

Table 3: GPU hours to generate 1:1 augmenting training data replica. SD involves downsampling for CIFAR-10.

| Dataset | R3GAN | EDM/EDM2 | CTM-EDM | SiD-EDM | SD |
|---|---|---|---|---|---|
| CIFAR-10 | 26.0s | 3102s | 58.3s | 39.1s | 6.8h |
| IN1K | - | 291h | - | 8.7h | 1290h |

## 5.2 Substituting Augmentation Results

We next consider evaluate DAR-GDA in a full data replacement scenario, where hypotheses are trained solely on synthetic samples equivalent in volume and label distribution to the original training set. Tables 4 and 5 report the classification performance for CIFAR-10 and ImageNet-1K, respectively. While DAR-GDA consistently boosts accuracy over baseline synthetic data, a performance gap to training on real data is observed, this drop being more significant on ImageNet-1K. Notably, on ImageNet-1K, DAR-GDA enables hypotheses trained on synthetic data to match the performance achieved with data from the original, non-distilled multi-step diffusion teacher model.

Table 5: ImageNet-1K classification accuracy with ResNet-50 and ViT-S/16, using training sets *substituted by* synthetic data equal to 1× the original dataset size on various generative models and adversarial-distillation methods under static versus dynamic data generation schemes.

| Generative Model | DAR progression | | | Top-1 Acc. | Top-5 Acc. |
|---|---|---|---|---|---|
| | Distill | Adv. align | Reweight | | |
| ResNet-50 classifier | | | | | |
| Real only | - | - | - | 76.37±0.03 | 92.86±0.08 |
| Diffusion Model [40] (FID=1.91) | ✗ | ✗ | ✗ | 66.41±0.47 | 86.58±0.35 |
| SIDA [121] FID=1.37 | ✗ | ✗ | ✗ | 66.22±0.50 | 86.30±0.42 |
| | ✔ | ✔ | ✗ | 66.42±0.59 | 86.54±0.38 |
| | ✔ | ✔ | ✔ | **66.50**±0.87 | **86.82**±0.66 |
| ViT-S/16 classifier | | | | | |
| Real only | - | - | - | 79.10±0.03 | 94.43±0.08 |
| Diffusion Model [40] (FID=1.91) | ✗ | ✗ | ✗ | 67.89±0.38 | 85.55±0.29 |
| SIDA [121] FID=1.37 | ✗ | ✗ | ✗ | 67.54±0.43 | 85.09±0.39 |
| | ✔ | ✔ | ✗ | 67.73±0.50 | 85.30±0.47 |
| | ✔ | ✔ | ✔ | **68.01**±0.64 | **85.89**±0.56 |

On CIFAR-10, SiDA-generated data significantly outperforms CTM data, despite both models being distilled from the same teacher. Remarkably, fully-synthetic GDA with SiDA with dynamic generation can surpass the performance of GDA training on the real dataset alone.

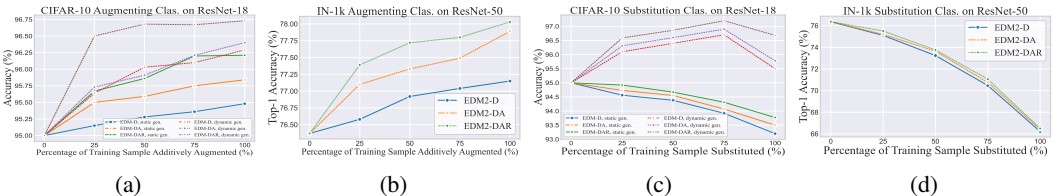

| (a) | (b) | (c) | (d) |

Figure 2: Classification accuracy of SiDA-distilled EDM/EDM2 models on CIFAR-10/IN-1K with varying synthetic dataset sizes (measured as a % of the original training set). Performance is shown under two schemes: additive augmentation (Figs. 2a, 2b) and full data substitution (Figs. 2c, 2d).

## 5.3 Further Empirical Studies

We present an ablation study on augmenting size in Fig. 2. For additive augmentation, performance increases with more synthetic data, and our method's components yield incremental gains. Conversely, under data substitution, performance generally declines with increasing synthetic data. This decline is more significant for ImageNet, where generative models face greater difficulty matching the true data distribution. Notably, on CIFAR-10, a dynamic generation schedule improves performance with up to 75% of the synthetic data replacement, where diversity from new per-epoch samples appears to offset the loss of real data. Additional empirical studies are presented in Appendix D.

## 6 Discussion and Conclusion

**Limitations and Ethical Considerations.** Our methodology concentrates on GDA itself, rather than on the development of new generative models. Hence, we leveraged state-of-the-art EDM diffusion models as the teacher. While this approach entails a high, one-time pre-training cost, our framework does not address this; rather, our focus is on making these powerful, pre-trained models highly efficient for low-latency applications like GDA. A related limitation is that our validation, while rigorous, was constrained to the two large-scale datasets for which these specific teacher weights are publicly available, though the observed patterns were consistent and supported our hypotheses. Ethically, our GDA framework contributes positively to content safety by enabling stronger and more reliable discriminative models for detecting and filtering harmful or NSFW material. Nevertheless, because the generator component is trained to replicate the visual characteristics of harmful content for the purpose of improving filtering, it inherently carries the capability to reproduce such material. To ensure responsible use, the trained generator should be securely stored to prevent misuse.

**Future Work.** Looking ahead, the core principles of DAR-GDA are fundamentally domain-agnostic. Our framework treats the teacher model as a "black box" for score predictions. This modularity allows the framework to be applied beyond vision, for instance, to emerging diffusion models in domains like protein [102] and molecule generation [12], simply by replacing the U-Net backbone with a domain-specific architecture. This is particularly promising for data-scarce fields like medicine (*e.g.*, MRI or X-ray synthesis), where a distilled student model could be shared to enable "pseudo data sharing" without violating patient confidentiality. Beyond new domains, the framework is applicable to a broader range of conditional tasks. For instance, in text-to-image generation, the discriminator component could be adapted to evaluate image quality and prompt alignment, *e.g.*, assessing $p(\text{image} \mid \text{prompt})$. The same principle applies to dense prediction tasks like segmentation, where the challenge shifts towards pseudo-labeling from foundation models or other forms of weak supervision to create the necessary training signals.

**Conclusion.** In this paper, we addressed two primary hindrances to the practical application of current diffusion models in GDA: suboptimal sampling efficiency and potential misalignment between generated and target data distributions. We introduced DAR-GDA, a three-stage synergistic framework where adversarial alignment (A) serves as a crucial bridge connecting model distillation (D)—to enhance sampling speed—and sample reweighting (R)—to correct distributional shifts. Our experiments demonstrated that the progressive integration of these DAR components leads to consistent improvements in GDA performance. Beyond the class-conditional models explored, this work may inspire the application of similar principled strategies to a wider array of diffusion-based generative models for effective data augmentation.

## Acknowledgments

M. Zhou acknowledges the support of a gift grant from Apple.

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

# A More Mathematical Discussions

## A.1 Detailed Deriavations for Eq. 11

By the law of large numbers, the empirical reweighted loss converges to its population expectation under $\mathcal{Q}_G$:

$$|\mathcal{S}_{\mathcal{Q}_G}|^{-1} \sum_{(\boldsymbol{x},y)\in\mathcal{S}_{\mathcal{Q}_G}} r_D(\boldsymbol{x})\,\ell(h(\boldsymbol{x}),y) \approx \mathbb{E}_{(\boldsymbol{x},y)\sim\mathcal{Q}_G}\big[r_D(\boldsymbol{x})\,\ell(h(\boldsymbol{x}),y)\big]. \tag{16}$$

Substituting the definition of the density ratio $r(\boldsymbol{x}) = p_{\text{data}}(\boldsymbol{x})/q_G(\boldsymbol{x})$ gives

$$\begin{aligned}
\mathbb{E}_{(\boldsymbol{x},y)\sim\mathcal{Q}_G}\big[r(\boldsymbol{x})\,\ell(h(\boldsymbol{x}),y)\big] &= \int r(\boldsymbol{x})\,\ell(h(\boldsymbol{x}),y)\,q_G(\boldsymbol{x},y)\,d\boldsymbol{x}\,dy \\
&= \int \frac{p_{\text{data}}(\boldsymbol{x})}{q_G(\boldsymbol{x})}\,\ell(h(\boldsymbol{x}),y)\,q_G(\boldsymbol{x},y)\,d\boldsymbol{x}\,dy \\
&= \int p_{\text{data}}(\boldsymbol{x})\,q_G(y\mid\boldsymbol{x})\,\ell(h(\boldsymbol{x}),y)\,d\boldsymbol{x}\,dy.
\end{aligned} \tag{17}$$

The term $p_{\text{data}}(\boldsymbol{x})$ adjusts for the marginal discrepancy between $p_{\text{data}}$ and $q_G$, ensuring that the contribution of each sample $\boldsymbol{x}$ in the expectation reflects its true data likelihood. Hence, Eq. (17) represents the unbiased risk with respect to the true data marginal:

$$\mathbb{E}_{(\boldsymbol{x},y)\sim\mathcal{Q}_G}\big[r(\boldsymbol{x})\,\ell(h(\boldsymbol{x}),y)\big] = \mathbb{E}_{\boldsymbol{x}\sim p_{\text{data}}(\boldsymbol{x})}\Big[\mathbb{E}_{y\sim q_G(y|\boldsymbol{x})}[\ell(h(\boldsymbol{x}),y)]\Big]. \tag{18}$$

Assume the generator is sufficiently well-trained based on the strong conditionally pre-trained teacher and is class faithful, *i.e.*, $(\boldsymbol{x},y) \sim \mathcal{Q}_G$, the inner expectation naturally approximates $\mathbb{E}_{y\sim p_{\text{data}}(y|\boldsymbol{x})}[\ell(h(\boldsymbol{x}),y)]$. Consequently, the reweighted risk converges to the true expected risk:

$$\mathbb{E}_{(\boldsymbol{x},y)\sim\mathcal{Q}_G}\big[r_D(\boldsymbol{x})\,\ell(h(\boldsymbol{x}),y)\big] \approx \mathbb{E}_{(\boldsymbol{x},y)\sim\mathcal{P}}[\ell(h(\boldsymbol{x}),y)]. \tag{19}$$

This establishes that the reweighting mechanism based on the discriminator-estimated ratio $r_D(\boldsymbol{x})$ corrects for the marginal distributional shift between $q_G(\boldsymbol{x})$ and $p_{\text{data}}(\boldsymbol{x})$, thereby aligning optimization on $\mathcal{Q}_G$ with learning on $\mathcal{P}$.

## A.2 Detailed Derivation for Eq.15

Let $(\Omega, \mathcal{F}, \mu)$ be a measurable space and assume $\mathcal{P}, \mathcal{Q}$ admit densities $p = \frac{d\mathcal{P}}{d\mu}$ and $q = \frac{d\mathcal{Q}}{d\mu}$ w.r.t. $\mu$.

Define the total variational distance:

$$\mathcal{D}_{\text{TV}}(\mathcal{P}\|\mathcal{Q}) = \tfrac{1}{2}\int_\Omega |p - q|\,d\mu = 1 - \int_\Omega \min\{p, q\}\,d\mu. \tag{20}$$

For the bounded loss $|\ell| \le L$,

$$\Big|\mathbb{E}_\mathcal{P}[\ell(h)] - \mathbb{E}_\mathcal{Q}[\ell(h)]\Big| = \Big|\int \ell(h(\boldsymbol{x}),y)\,d(\mathcal{P}-\mathcal{Q})\Big| \tag{21}$$

$$\le \int |\ell(h(\boldsymbol{x}),y)|\,\big|d(\mathcal{P}-\mathcal{Q})\big| \tag{22}$$

$$\le L\int \big|d(\mathcal{P}-\mathcal{Q})\big| \tag{23}$$

$$= 2L\,\mathcal{D}_{\text{TV}}(\mathcal{P}\|\mathcal{Q}). \tag{24}$$

By Pinsker's inequality [13], for any $\mathcal{R}, \mathcal{S}$ with densities $r, s$,

$$\mathcal{D}_{\text{TV}}(\mathcal{R}\|\mathcal{S}) \le \sqrt{\tfrac{1}{2}\,D_{\text{KL}}(\mathcal{R}\|\mathcal{S})}. \tag{25}$$

Equivalently, $\mathcal{D}_{\text{KL}}(\mathcal{R}\|\mathcal{S}) \ge 2\,\mathcal{D}_{\text{TV}}(\mathcal{R}\|\mathcal{S})^2$.

Fix $\pi \in (0,1)$ and set the mixture $\mathcal{M} = \pi\mathcal{P} + (1-\pi)\mathcal{Q}$ with density $m = \pi p + (1-\pi)q$. Define the $\pi$-Jenson-Shannon divergence

$$D_{\text{JS},\pi}(\mathcal{P}\|\mathcal{Q}) = \pi\,D_{\text{KL}}(\mathcal{P}\|\mathcal{M}) + (1-\pi)\,D_{\text{KL}}(\mathcal{Q}\|\mathcal{M}). \tag{26}$$

Applying Pinsker's to each KL term and then expressing the two TVs through $\mathcal{D}_{\text{TV}}(\mathcal{P},\mathcal{Q})$:

$$\begin{aligned}
\mathcal{D}_{\text{TV}}(\mathcal{P}\|\mathcal{M}) &= \tfrac{1}{2}\int |p - m|\,d\mu = \tfrac{1}{2}\int |(1-\pi)(p-q)|\,d\mu = (1-\pi)\,\mathcal{D}_{\text{TV}}(\mathcal{P},\mathcal{Q}), \\
\mathcal{D}_{\text{TV}}(\mathcal{Q}\|\mathcal{M}) &= \tfrac{1}{2}\int |q - m|\,d\mu = \tfrac{1}{2}\int |\pi(q-p)|\,d\mu = \pi\,\mathcal{D}_{\text{TV}}(\mathcal{P},\mathcal{Q}).
\end{aligned} \tag{27}$$

Hence,

$$D_{\mathrm{JS},\pi}(\mathcal{P}\|\mathcal{Q}) \geq \pi \cdot 2\,\mathcal{D}_{\mathrm{TV}}(\mathcal{P}\|\mathcal{M})^2 + (1-\pi) \cdot 2\,\mathcal{D}_{\mathrm{TV}}(\mathcal{Q}\|\mathcal{M})^2 \tag{28}$$

$$= 2\Big(\pi(1-\pi)^2 + (1-\pi)\pi^2\Big)\mathcal{D}_{\mathrm{TV}}(\mathcal{P}\|\mathcal{Q})^2 \tag{29}$$

$$= 2\,\pi(1-\pi)\,\mathcal{D}_{\mathrm{TV}}(\mathcal{P}\|\mathcal{Q})^2. \tag{30}$$

For the standard Jensen-Shannon divergence, *i.e.*, $\pi = \frac{1}{2}$,

$$D_{\mathrm{JS}}(\mathcal{P}\|\mathcal{Q}) \geq \tfrac{1}{2}\,\mathcal{D}_{\mathrm{TV}}(\mathcal{P}\|\mathcal{Q})^2 \tag{31}$$

$$\mathcal{D}_{\mathrm{TV}}(\mathcal{P}\|\mathcal{Q}) \leq \sqrt{2\,D_{\mathrm{JS}}(\mathcal{P}\|\mathcal{Q})}. \tag{32}$$

Combining the results, we have

$$\Big|\mathbb{E}_{\mathcal{P}}[\ell(h)] - \mathbb{E}_{\mathcal{Q}}[\ell(h)]\Big| \leq 2L\,\mathcal{D}_{\mathrm{TV}}(\mathcal{P}\|\mathcal{Q}) \leq 2L\,\sqrt{2\,D_{\mathrm{JS}}(\mathcal{P}\|\mathcal{Q})}. \tag{33}$$

Lastly, since any quantity $x$ is trivially upper-bounded by its absolute value, *i.e.*, $x \leq |x|$, we relate Eq. 33 to the first summand in Eq. 15 by $\Big|\mathbb{E}_{\mathcal{P}}[\ell(h)] - \mathbb{E}_{\mathcal{Q}}[\ell(h)]\Big| \geq \mathbb{E}_{\mathcal{P}}[\ell(h)] - \mathbb{E}_{\mathcal{Q}}[\ell(h)]$, giving a valid upper bound for the non-absolute risk difference.

# B  Algorithmic Description of GDA-DAR

We summarize the overall training scheme in Algorithm 1, which illustrates a dynamic, alternating adversarial optimization. For conceptual clarity, the algorithm presents the discriminator update as a distinct step, corresponding to an architecture where $D_\eta$ is a standalone network [43]. However, as detailed in Section 4.3, alternative formulations often integrate the discriminator into the student's score network (*i.e.*, the "fake" U-Net) to leverage a shared encoder [119, 121]. In such cases, the parameters $\eta$ are subsumed by the score network parameters $\psi$; consequently, the separate discriminator step is omitted, and $\psi$ is updated jointly to simultaneously maximize the adversarial value function and minimize the score matching error on the perturbed generated data.

# C  Experimental Details

## C.1  Details of the Datasets

CIFAR-10 consists of 60,000 RGB images, each with a resolution of $32 \times 32$ pixels. The dataset is evenly distributed across 10 distinct classes, with each class containing 6,000 images. By default, these are split into 5,000 training samples and 1,000 test samples per class, offering a balanced and computationally efficient benchmark for evaluating machine learning algorithms.

ImageNet-1K is a dataset comprising 1.28 million images labeled across 1,000 object categories. Each image is annotated with one or more class labels, enabling comprehensive studies of object recognition in diverse real-world scenarios.

## C.2  Details of the Metrics

We assess model performance based on accuracy. For ImageNet-1K, which contains 1,000 possible classes, we report both Top-1 and Top-5 accuracies.

We evaluate the generative models based on the Fréchet Inception Distance (FID) [30], a widely used metric to evaluate the quality of images generated by generative models. This is based on the distance between the feature distributions of real and generated images in the latent space of a pretrained Inception network.

Table 3 details the GPU hours required to generate the augmenting dataset. For CIFAR-10, computations were performed on a single 80GB-A100 GPU. For other datasets, we utilized eight 80GB-A100 GPUs, employing the maximum possible batch size that is a power of two.

**Algorithm 1** Generative data augmentation in supervised training with GDA-DAR framework.

---

**Input:** Teacher diffusion model $\epsilon_\phi$; real dataset $\mathcal{S}_\mathcal{P}$; hyperparameters: learning rates $\alpha_\theta, \alpha_\eta, \alpha_h$, loss weights $\lambda_1, \lambda_2, \lambda_{\text{aug}}$, truncation threshold $\gamma$; number of training iterations $N_{\text{gen}}, N_{\text{clf}}$.

---

**Phase 1: Adversarial Score Distillation (D & A stages)**
Initialize student generator $G_\theta$ and auxiliary score network $\epsilon_\psi$ from teacher $\epsilon_\phi$.
Initialize discriminator $D_\eta$.
**for** $i = 1, \ldots, N_{\text{gen}}$ **do**
    Sample real batch $\mathbf{x} \sim \mathcal{S}_\mathcal{P}$ and noise $\mathbf{z} \sim \mathcal{N}(\mathbf{0}, \mathbf{I})$.
    *// Discriminator Update*
    Compute adversarial value $\mathcal{V}_{\text{adv}}(\eta)$ on real $\mathbf{x}$ and synthetic $G_\theta(\mathbf{z})$ by Eq. 13.
    Update discriminator: $\eta \leftarrow \eta - \alpha_\eta \nabla_\eta (-\mathcal{V}_{\text{adv}}(\eta))$.
    *// Distillation*
    Construct perturbed synthetic samples: $\boldsymbol{y}_t = \alpha_t G_\theta(\boldsymbol{z}) + \sigma_t \boldsymbol{\epsilon}_t$ where $\boldsymbol{z}, \boldsymbol{\epsilon}_t \sim \mathcal{N}(\mathbf{0}, \mathbf{I}), t \sim p(t)$
    Compute fake score loss: $\mathcal{L}^{(\text{fake score})}(\psi) = \mathbb{E}_{\boldsymbol{z}, \boldsymbol{\epsilon}_t, t}\big[\|\boldsymbol{\epsilon}_t - \epsilon_\psi(\boldsymbol{y}_t, t)\|_2^2\big]$.
    Update score network: $\psi \leftarrow \psi - \alpha_\psi \nabla_\psi \mathcal{L}^{(\text{fake score})}(\psi)$.
    Compute score distillation loss $\mathcal{L}^{(\text{SD})}(\theta)$ (Eq. 4).
    Compute generator adversarial loss $\mathcal{L}_G^{(\text{adv.})}(\theta) = -\mathbb{E}_{\mathbf{z}}[\log D_\eta(G_\theta(\mathbf{z}))]$.
    Update generator: $\theta \leftarrow \theta - \alpha_\theta \nabla_\theta \big(\lambda_1 \mathcal{L}^{(\text{SD})}(\theta) + \lambda_2 \mathcal{L}_G^{(\text{adv.})}(\theta)\big)$.
**end for**

---

**Phase 2: Reweighted Classifier Training (R stage)**
Initialize discriminative hypothesis (classifier) $h$.
Freeze parameters $\theta$ and $\eta$.
**for** $i = 1, \ldots, N_{\text{clf}}$ **do**
    *// Real Data Path*
    Sample real batch $\{\boldsymbol{x}_j, y_j\}_{j=1}^B \sim \mathcal{S}_\mathcal{P}$.
    Compute real loss $\mathcal{L}^{(\text{real})} = \frac{1}{B} \sum_{j=1}^B \ell(h(\boldsymbol{x}_j), y_j)$.
    *// Synthetic Augmentation Path*
    Let synthetic batch size $B_{\text{aug}} = \lfloor \lambda_{\text{aug}} B \rfloor$.
    Sample noise batch $\{\boldsymbol{z}_s, y_s\}_{j=1}^{B_{\text{aug}}} \sim \mathcal{N}(0, \mathbf{I}) \times \text{Cat}(\mathcal{Y})$.
    Generate synthetic batch $\{\hat{\boldsymbol{x}}_s\}_{j=1}^{B_{\text{aug}}} = G_\theta(\{\boldsymbol{z}_s\}_{j=1}^{B_{\text{aug}}})$.
    *// Compute Importance Weights for synthetic batch*
    **for** $j = 1, \ldots, B_{\text{aug}}$ **do**
        Compute density ratio $r_j = D_\eta(\hat{\boldsymbol{x}}_{s,j})/(1 - D_\eta(\hat{\boldsymbol{x}}_{s,j}))$.
        Truncate weight $\tilde{r}_j = \min(r_j, \gamma)$.
    **end for**
    **if** self_norm **then**
        Self-normalize: $\tilde{r}_j \leftarrow \bar{r}_j \big/ \sum_{k=1}^{B_{\text{aug}}} \tilde{r}_k \quad$ for $j = 1, \ldots, B_{\text{aug}}$
    **end if**
    Compute reweighted loss $\mathcal{L}^{(\text{reweight})} = \sum_{j=1}^{B_{\text{aug}}} \tilde{r}_j \cdot \ell(h(\hat{\boldsymbol{x}}_{s,j}), y_{s,j})$.
    *// Combined ppdate of the hypothesis $h$*
    Compute total loss $\mathcal{L}_{\text{total}} = \mathcal{L}^{(\text{real})} + \mathcal{L}^{(\text{reweight})}$.
    Update hypothesis: $h \leftarrow h - \alpha_h \nabla_h \mathcal{L}_{\text{total}}$.
**end for**

---

**Output:** Trained hypothesis $h$.

---

## C.3   Details of the Implementation

For CIFAR-10, we set $\alpha = 1.2$ for SiDA and train using the Adam optimizer with a learning rate of $1 \times 10^{-5}$. CTM is trained with a batch size of 128 for 256 steps per batch, using a student learning rate of $3 \times 10^{-4}$ and a discriminator learning rate of $2 \times 10^{-3}$. For ImageNet-1K, we distill EDM2-XXL using SiDA across 8 A100-80GB GPUs with $\alpha = 1.0$, per-GPU batch size 64, and gradient accumulation every 128 iterations, using Adam with learning rate $5 \times 10^{-5}$. CIFAR-10

models are trained for 300 epochs with a batch size of 128 using momentum SGD (learning rate 0.1). Hyperparameters $\lambda_1$ and $\lambda_2$ for adversarial alignment follow prior work.

For ImageNet-1K, ResNet-50 is trained for 90 epochs with batch size 4096 and initial learning rate 1.6, while ViT-S/16 is trained for 300 epochs with batch size 1024, using AdamW [58] and initial learning rate $3 \times 10^{-3}$. Self-normalization is applied with $\gamma = 1$ to obtain $r(x)$ for reweighting. Diffusion outputs are generated at $32 \times 32$ resolution for CIFAR-10 and $512 \times 512$ for ImageNet, with the latter downsampled to $224 \times 224$ before classification to match baseline protocols.

### C.3.1 Classifier Training

The hyperparameters used for training the hypotheses, *i.e.*, the classification models, on the CIFAR-10 and ImageNet-1K datasets are detailed in Tables A1 and A2, respectively. Models are trained using a Stochastic Gradient Descent (SGD) optimizer across 90 epochs. The learning rate is programmed to initially ramp up from 0.0 to 0.4 over the first five epochs, followed by decrements of 0.1 at the 30th, 60th, and 80th epochs.

For CIFAR-10 training, we utilize standard augmentation techniques including padding each image by 4 pixels on all sides, then performing a random $32 \times 32$ crop from either the padded image or its horizontal flip, following the methods described in [28]. For ImageNet-1K, the models undergo training using a $224 \times 224$ random crop, applied with bilinear interpolation, from either the original image or its horizontal flip.

The detailed recipes of the hypothesis training are reported in Table A1 and Table A2 respectively.

Table A1: Training recipes for ResNet-18 and VGG-16 on CIFAR-10.

| Model | ResNet-18 | VGG-16 |
|---|---|---|
| Batch size | 128 | 128 |
| Epochs | 300 | 300 |
| Optimizer | Momentum SGD | Momentum SGD |
| Learning rate | 0.1 | 0.1 |
| LR scheduler | CosineAnnealingLR | CosineAnnealingLR |
| Nesterov | True | True |
| Weight decay | 5e-4 | 5e-4 |

Table A2: Training recipes for ResNet-50 and ViT-S/16 on ImageNet-1K.

| Model | ResNet-50 | ViT-S/16 |
|---|---|---|
| Batch size | 4096 | 1024 |
| Epochs | 130 | 300 |
| Optimizer | Momentum SGD | AdamW |
| Learning rate | 1.6 | 0.003 |
| LR scheduler | CosineannealingLR | CosineAnnealingLR |
| Weight decay | 1e-4 | 0.3 |
| Warmup epochs | 5 | 30 |
| Label smoothing | 0 | 0.11 |
| Mixup probability | 0 | 0.2 |
| Cutmix $\alpha$ | 0 | 1.0 |

### C.3.2 Generative models training

For the base diffusion model, we adopt the public weight for EDM Diffusion on CIFAR-10 and EDM2 on ImageNet-1K for their state-of-the-art performance.

For experiments on the CIFAR-10 dataset (see Tables 1 and 4), we set $\alpha = 1.2$ for SiDA and train using the Adam optimizer with a learning rate of 1e-5. The batch size is set to 256, and gradient accumulation is performed every iteration (i.e., accumulation round = 1). We do not use mixed precision training (fp16) for SiDA on CIFAR-10. Dropout and data augmentation settings follow those used in EDM for CIFAR-10. All other SiDA-specific hyperparameters are kept consistent with the original implementation as described in [121].

For CTM, CTM is trained with batch size 128 for 256 steps per batch, using a student learning rate of $3 \times 10^{-4}$ and a discriminator learning rate of $2 \times 10^{-3}$ Mixed precision training (fp16) was employed to match the original setup. The exponential moving average (EMA) decay rate is 0.999. Other CTM-specific parameters adhere to the settings reported in [43].

On the ImageNet-1K dataset, we distilled EDM2-XXL with a batch size of 2048 and gradient accumulation performed every 128 iterations. train using the Adam optimizer with a learning rate of 5e-5. Mixed precision training (fp16) was used for SiDA on this dataset. Dropout parameters follow those of EDM2 on ImageNet-1K, and all other SiDA-specific hyperparameters remain consistent with those in the original work [121].

### C.3.3 Training Procedure

In our experiments on the CIFAR-10 dataset (shown in Tables 1 and 4), we investigated two distinct strategies for generating synthetic data: the static approach and the dynamic approach. For the static approach, we created 50,000 synthetic samples, equivalent in size to the original dataset, which remained consistent throughout the entire training process. In contrast, the dynamic approach involved generating a new set of 50,000 synthetic samples at the start of each epoch, ensuring that the synthetic dataset used for training was entirely refreshed every epoch. For the experiments on the ImageNet-1K dataset, we generated 1.28 million synthetic samples for training, which is equivalent to the size of the original dataset. Given the large scale of ImageNet-1K, regenerating the synthetic dataset at each epoch would be computationally expensive and time-consuming. Therefore, we opted for the static approach.

### C.3.4 Implementation Details and Notes

While our framework comprises three stages, its implementation is manageable due to its modular design. The (D), (A), and (R) stages are sequential and decoupled. As noted in relevant work [121, 43], stability is observed in the adversarial distillation. This can be attributed to the design of initializing the student with a pre-trained teacher, which bypasses the unstable early stages of typical GAN training. Hence, for the student generator and discriminator training phase, a single, fixed set of distillation hyperparameters was used across all datasets. For the final classifier training (R) stage, we simply follow standard, existing recipes from the literature. In addition,

The primary tuning consideration *i)* balancing the loss weights $\lambda_1$ and $\lambda_2$ remain at a comparable magnitude; and *ii)* considering truncation threshold $\gamma$ and the use of self-normalization to achieve a favorable bias-variance trade-off in the estimated importance ratios.

Additionally, a key in-process indicator of a successful D+A phase is the student generator's own performance. We (and others [121, 43, 110, 59]) observe that the student's FID score remains stable and often improves beyond that of the original teacher. This provides direct evidence that the discriminator is providing high-quality gradients and has successfully captured the real-versus-generated distribution gap. We recommend users monitor the student's generation quality as a primary check for a healthy run.

## D Additional Experiment

### D.1 Ablation Study of Variance Reduction Techniques

We ablate on the two variance-reducing techniques used, namely *self-normalization* and *trucation*, with respect to the hyperparameter that controls the upper bound of $\gamma$ in Table A3. Our observations indicate that removing self-normalization generally leads to a decrease in classification performance. This decrease is particularly significant on ImageNet-1K, likely because weight variance can be greater with such a massive dataset. Furthermore, we observe that $\gamma = 1$ appears to be the optimal upper bound for truncation. At this value, a synthetic image contributes equally to a real image in the loss computation and subsequent optimization.

Table A3: Classification accuracies (top-1) under different weight variance reduction techniques for additive, static augmentation using EDM/EDM2-SiD distillation.

| Self-norm | $\gamma$ | CIFAR-10 | | ImageNet-1K | |
| --- | --- | --- | --- | --- | --- |
| | | ResNet-18 | VGG-16 | ResNet-50 | ViT-S/16 |
| ✗ | 0.5 | 95.30 | 94.08 | 77.09 | 80.05 |
| ✗ | 1 | 96.15 | 94.60 | 77.75 | 80.90 |
| ✗ | 2 | 95.72 | 94.10 | 77.23 | 80.12 |
| ✔ | 0.5 | 95.70 | 94.37 | 77.73 | 80.47 |
| ✔ | 1 | 96.21 | 94.64 | 78.03 | 81.17 |
| ✔ | 2 | 95.86 | 94.40 | 77.64 | 80.68 |

In addition, we conducted an empirical study reporting the sample variance of the weights for 100,000 generated ImageNet samples. As shown in Table A4, both truncation and self-normalization progressively reduce the variance.

Table A4: Sample variance of importance weights $r(x)$ for 100,000 generated ImageNet samples, under different truncation ($\gamma$) and self-normalization (SN) schemes.

| Truncation $\gamma$ | Variance of $r(x)$ (no SN) | Variance after SN |
| --- | --- | --- |
| $\infty$ (no clip) | 1.21 | 0.42 |
| 2.0 | 0.37 | 0.12 |
| 1.0 | 0.18 | 0.05 |
| 0.5 | 0.07 | 0.02 |

## D.2 Discriminator Noising Scheme

When employing the discriminator as the encoder for the student generator's backbone, it's necessary to transform the clean input, $x_{\text{clean}}$, into a noisy version, $x_{\text{noisy}}$. This is achieved by corrupting $x_{\text{clean}}$ with a predetermined level of noise corresponding to a specific timestep $t_0$. The choice of this noising scheme, specifically the selection of $t_0$, can influence the features learned by the discriminator-encoder and subsequently the performance of the student generator. Please note that the original $p(t)$ is used for the computation of $\mathcal{L}^{(\text{fake score})}(\psi)$.

We investigated several approaches for selecting $t_0$, as detailed in Table A5. We observe that the uniform sampling strategy is the best among all ablated ones with $t_0 = 0.1T$ having comparable performance on both datasets. The approach of treating the shared encoder as the discriminator offers a trade-off: it obviates the need to tune the noising scheme, while avoiding the potential complexities and additional computational overhead associated with employing a separate, dedicated discriminator network.

Table A5: Classification accuracies (top-1) under different discriminator noising schemes for static additive augmentation using EDM/EDM2-SiD distillation.

| $\pi$ | CIFAR-10 | | ImageNet-1K | |
| --- | --- | --- | --- | --- |
| | ResNet-18 | VGG-16 | ResNet-50 | ViT-S/16 |
| $\delta(0.05T)$ | 96.08 | 94.48 | 77.87 | 81.02 |
| $\delta(0.1T)$ | 96.18 | 94.66 | 77.89 | 81.10 |
| $\delta(0.5T)$ | 95.33 | 94.07 | 77.23 | 80.51 |
| $\mathcal{U}(0, 0.5T)$ | 96.21 | 94.64 | 78.03 | 81.17 |

## D.3 Noisy and Imbalanced Setting

While our initial goal was to focus on challenges like distribution mismatch and sampling cost in GDA, we further investigated how our approach would be suitable for these real-world scenarios such as on imbalanced and noisy data.

**Class Imbalance** We conducted experiments on CIFAR-10-LT [16] with an imbalance factor of 10 and 100 using the sa, me training protocol as in the balanced-data setting the dynamic scheme, with

ResNet-18 as the backbone in Tables A6 and A7 respectively. We also included a long-tailed-based method, SURE [55], for comparison. The evaluation tracks the progressive impact of each stage in our DAR-GDA pipeline (+D), (+A), and Reweighting (+R) with SiD [122]-distilled EDM [39] diffusion model. This performance likely stems from our method's ability to leverage the strong generative priors of the diffusion model teacher to populate tail classes, a different paradigm from methods like SURE, which are designed to learn directly from the imbalanced data.

Table A6: Results on CIFAR-10-LT with Imbalance Factor = 10.

| Method | Augment | Substitute |
|---|---|---|
| Baseline | 86.75 | 86.75 |
| SURE [55] | 95.15 | 95.15 |
| EDM-SID (+D) | 94.88 | 94.47 |
| EDM-SID (+D+A) | 95.53 | 94.98 |
| EDM-SID (+D+A+R) | **95.78** | 95.02 |

Table A7: Results on CIFAR-10-LT with Imbalance Factor = 100.

| Method | Augment | Substitute |
|---|---|---|
| Baseline | 69.76 | 69.76 |
| SURE [55] | 86.75 | 86.75 |
| EDM-SID (+D) | 91.45 | 91.02 |
| EDM-SID (+D+A) | 91.84 | 91.35 |
| EDM-SID (+D+A+R) | **92.01** | 91.64 |

**Noisy Label** We also conducted an experiment on the CIFAR-10-N dataset [103] using the "aggre" noise labels to simulate a real-world scenario with untrusted data. We used a ResNet-18 backbone with the augmenting data protocol. We compare our result with a noisy-label learning method, ProMix [104] in Table A8.

Table A8: Results on CIFAR-10-N with ResNet-18.

| Method | Static | Dynamic |
|---|---|---|
| Baseline | 87.77 | 87.77 |
| ProMix [104] | **97.65** | **97.65** |
| EDM-SID (+D) | 93.48 | 95.79 |
| EDM-SID (+D+A) | 93.80 | 96.26 |
| EDM-SID (+D+A+R) | 93.81 | 96.30 |

Our method contains no explicit mechanism for label correction; the discriminator is designed to down-weight visually atypical samples, not to detect label noise. Consequently, while the strong generative prior provides a robust foundation, the performance does not surpass that of specialist methods like ProMix, which are explicitly architected to handle label noise.

## D.4    Comparisons with Non-generative Data Augmentation Methods

Our primary work focused on challenges within the GDA paradigm. To provide a broader context, we conducted additional experiments comparing our DAR-GDA framework against two widely-used non-generative data augmentation baselines: AutoAugment [14] and RandAugment [15].

Table A9: Comparison with non-generative augmentation on CIFAR-10 with ResNet-18.

| Method | Static | Dynamic |
|---|---|---|
| Real-only | 95.00 | 95.00 |
| AutoAugment [14] | 95.85 | 95.85 |
| RandAugment [15] | 95.79 | 95.79 |
| EDM-SID (+D) | 95.48 | 96.29 |
| EDM-SID (+D+A) | 95.84 | 96.40 |
| EDM-SID (+D+A+R) | 96.21 | 96.73 |
| EDM-SID (+D+A+R) + RandAugment | **96.48** | **96.97** |

Table A10: Comparison with non-generative augmentation on ImageNet-1K with ResNet-50.

| Method | Top-1 |
|---|---|
| Real-only | 76.37 |
| AutoAugment [14] | 77.62 |
| RandAugment [15] | 77.65 |
| EDM-SID (+D) | 77.15 |
| EDM-SID (+D+A) | 77.89 |
| EDM-SID (+D+A+R) | 78.03 |
| EDM-SID (+D+A+R) + RandAugment | **78.29** |

We notice that our full pipeline (DAR-GDA +D+A+R) outperforms both AutoAugment and RandAugment. Furthermore, we highlight that our generative approach is complementary to these traditional, non-generative data augmentation techniques. These transforms can be applied to synthetic images just as they are to real ones, further diversifying the training set. To demonstrate this, we applied RandAugment on top of our generated data, which yields further improvement.

### D.5 On FFHQ Dataset

To further validate our framework's robustness, we conducted experiments on a gender classification task using the FFHQ 64x64 dataset [41]. Since the EDM teacher and our student generator are unconditional, we first created a high-quality labeled dataset by using an external classifier of ResNet-50 pretrained on Celeb-A to generate raw predictions, which were then human-verified. We then applied our "static" scheme with results shown in Table A11. We observe monotonic performance gains as each stage of the DAR pipeline is applied.

Table A11: Gender classification accuracy on FFHQ 64x64. The baseline was trained on real data with human-verified pseudo-labels.

| Method | ResNet-18 | VGG-16 |
|---|---|---|
| Real-only | 94.08 | 93.10 |
| EDM-SID (+D) | 94.20 | 93.25 |
| EDM-SID (+D+A) | 94.63 | 93.62 |
| EDM-SID (+D+A+R) | **94.87** | **93.70** |

## E Computational Complexity Analysis

We perform a formal analysis of the computational complexity for our framework. We define the following computational costs:

$T$:      The number of denoising steps in the teacher sampler, *e.g.*, $T \approx 1000$.

$c_f$:      The wall-clock cost of one forward pass of the diffusion model backbone (assumed the same for the teacher and the one-step student).

$N$:        The total number of synthetic instances to be generated.

$C_{\text{distill}}$:        The one-time, upfront cost of distilling the teacher into the student generator.

$C_{\text{pretrain}}$:        The cost of pre-training the teacher model, which is shared/sunk for both methods and cancels out in the comparison.

The core of our efficiency gain comes from reducing the per-sample generation cost from the $\mathcal{O}(T)$ complexity of iterative samplers, like DDPM [31] or DDIM [82], to an $\mathcal{O}(1)$ complexity. Table A12 contrasts these approaches.

Table A12: Comparison of sampling time complexity and theoretical speed-up.

| Sampler | # Reverse Steps | Asymptotic Cost | Theoretical Speed-up (vs. 1000-step DDPM) |
|---|---|---|---|
| DDPM | 1000 | $\mathcal{O}(T)$ | Baseline ($1\times$) |
| DDIM (fast) | 50 | $\mathcal{O}(T)$ | $\approx 20\times$ |
| One-step distilled | 1 | $\mathcal{O}(1)$ | $\approx 1000\times$ |

With these variables, we can compare the total cost of generating $N$ samples using a naive iterative approach versus our DAR-GDA framework. The costs are summarized in Table A13.

Table A13: Computational cost comparison for generating $N$ synthetic samples.

| Pipeline | Total Cost Equation |
|---|---|
| Naive Diffusion GDA | $C_{\text{naive}} = N \cdot T \cdot c_f$ |
| DAR-GDA | $C_{\text{DAR}} = C_{\text{distill}} + N \cdot c_f$ |

Our framework (DAR-GDA) is faster than the naive approach whenever $C_{\text{DAR}} < C_{\text{naive}}$, which implies:

$$C_{\text{distill}} + N c_f < N T c_f \quad \implies \quad C_{\text{distill}} < N c_f (T - 1) \tag{34}$$

We can estimate the one-time distillation cost as $C_{\text{distill}} = P \cdot N \cdot k \cdot c_f$, where $P$ is the number of data passes (epochs) during distillation, $N$ is the number of data samples in the training set, and $k$ is the number of model evaluations per sample per pass.

Empirically, recent works [121, 43] suggest that one-step distillation converges in at most $P \leq 150$ passes, with $k = 3$, *i.e.*, teacher, fake, and student score network forwards. Assuming a teacher with $T = 1000$ steps, the ratio of distillation cost to savings is:

$$\frac{C_{\text{distill}}}{N c_f (T - 1)} = \frac{P N k c_f}{N c_f (T - 1)} \approx \frac{P k}{T - 1} \approx \frac{150 \cdot 3}{999} \approx 0.45 < 1 \tag{35}$$

Thus, the inequality is comfortably satisfied. Furthermore, if synthetic data is dynamically refreshed each epoch , *i.e.*, the "dynamic" scheme, the savings compound. In that case, the effective $N$ becomes $N \cdot E$ (where $E$ is the number of classifier training epochs), making the efficiency gain of DAR-GDA even more significant.

