# OpenReview forum: "Generative Data Augmentation via Diffusion Distillation, Adversarial Alignment, and Importance Reweighting"
_NeurIPS.cc/2025/Conference — NeurIPS 2025 poster_

### Official Review · Reviewer_BfFU · 2025-06-20

**Clarity:** 2
**Significance:** 3
**Originality:** 2
**Rating:** 5
**Confidence:** 3

**Summary:**

This submission proposes an interesting idea of efficiently using Probabilistic Diffusion Models in data augmentation of discriminative models. A naïve use of generative synthetic sample-label pairs with diffusion models has limitations of ‘slow generation’ and ‘synthetic-real distribution gap’. The key idea here is to use score-distillation (a known concept) of the diffusion model to estimate a student model that can generate image in a single denoising step. The student generative model is also guided within a GAN-type framework that aims at aligning the synthetic data distribution to real data (available as training data of the discriminative model). The discriminator of the GAN-setup is also used to estimate a ‘probability ratio’ used as a measure of typicality of the synthetic sample in the real data distribution. This ratio is used to re-weight the synthetic samples in the training of the discriminative model for the eventual classification task.
The method is tested for CIFAR-10 and ImageNet-1K, showing favourable results against direct use of synthetic data augmentation, and demonstrating general performance improvements with GAN-type setup and reweighting along with distillation.

**Questions:**

-	In authors’ view, how critical is the availability of the teacher generative model on the dataset of the eventual discriminative problem?
-	Why would the dynamic strategy still lead to a considerable gain over static generation in Table 4 when the training data is completely substituted with the synthetic data? There does not seem to be any additional feedback mechanism to alter the generation quality in that mode. Reviewer expected the numbers to be close to the static generation in Table 4. It will help if the authors can explain.
-	It seems the ratio r(x) could also be used to filter synthetic samples falling far outside p_data. Could the authors comment on this thought.
-	Do the authors plan to also make the source code public? Since there are many details in the approach that can be better understood through the code, it will be helpful to release the implementation for replication.
- The distillation and its embedding in GAN framework seems to be tailored for Markov chain based diffusion models, e.g., DDPM. Could the authors comment on what changes might be needed for extending the method to non-Markov models, e.g., DDIM. Also, how the compute gain of the proposed synthetic data generation would fare in that case.

**Ethical Concerns:**

["NO or VERY MINOR ethics concerns only"]

**Final Justification:**

Reviewer concerns are addressed and queries well-answered.

**Limitations:**

Authors could also mention in the paper about the potential negative societal impact regarding efficient deepfake generation with their method, and generation of NSFW images.

**Paper Formatting Concerns:**

None noticed.

**Quality:**

3

**Strengths And Weaknesses:**

Strengths:

-	The key idea of the paper regarding adversarially aligning a distilled model and using the discriminator for estimating importance weights, is interesting.
-	Experiments and results systematically establish that the overall approach works and its constituents progressively add to the performance of the method.
-	Details and discussion about results is comprehensive.

Weaknesses:

-	The paper suffers a bit from poor presentation. Whereas the method is interesting, it’s a bit hard to comprehend because no schematics, pseudo-code or algorithmic flow is presented.
-	There are plenty of minor errors and typos, that further complicate understanding the paper. Line numbers of a few are mentioned: L46, L59, L140, L141, L217, L300.
-	Results are only provided on two data sets.
-	It seems that the approach has a pre-requisite that a teacher generative diffusion model for the classification problem’s training data is available for the method to work.

---

> ### Author Rebuttal · Authors · 2025-07-31
>
> Thank you for your detailed and thoughtful review of our paper. We are very encouraged that you found the core idea of our work to be “**interesting**” and appreciated that our experiments "**systematically** establish that the overall approach works” with “**detailed discussion**”.
>
> Your feedback regarding the paper's presentation and clarity is particularly valuable. You have identified several key areas for improvement that we are eager to address. Below, we respond to each of your points and questions and outline our plan for revision.
>
> **[W1, W2] On Presentation, Clarity, and Typos**
>
> We are very grateful for this direct and actionable feedback. We take the presentation and clarity issue very seriously.
>
> **Action:** To fundamentally improve the paper's clarity and presentation, we will make two major additions to the revised manuscript:
>
> 1. **A new figure in the method section** to provide a step-by-step schematic of our proposed framework
> 2. **A new algorithm box in the appendix** presenting the complete training procedure in pseudocode, detailing the inputs, outputs, initialization, and calculation with respect to equations.
> 3. We have also done a **thorough proofreading** of the entire manuscript to correct all the errors you noted and any others we find. We are confident these changes will dramatically improve the paper's clarity. Delivering a well-written paper is our top priority in the revision.
>
> **[W3] On Limited Datasets**
>
> Our primary goal for this foundational work was to rigorously validate our three-stage framework on a well-established and understood frontier: large-scale image datasets. We believed it was essential to first prove its stability, effectiveness, and scalability in a domain with standardized benchmarks and felt this was the most scientifically sound way to build a strong foundation that could be reliably extended to other domains in the future.
> Nevertheless, to demonstrate the versatility and robustness of our framework beyond these initial datasets, we have conducted additional evaluations on the FFHQ dataset (64x64), where we apply our DAR stages progressively (+D, +A, +R) on the EDM-SID model to statically augment the classifier for a gender classification task:
>
> **Gender classification on FFHQ 64×64**
> |Method| ResNet-18| VGG-16|
> |---|---|---|
> |Baseline|94.08|93.10|
> |EDM-SID (+D)|94.20|93.25|
> |EDM-SID (+D+A)|94.63| 93.62|
> |EDM-SID (+D+A+R)|**94.87**|**93.70**|
>
> Here, we observe the same monotonic gains from the D-A-R progression, as seen on CIFAR/ImageNet. Furthermore, at the suggestion of Reviewer W9GD, we have added a comprehensive evaluation on challenging real-world scenarios, including long-tail and noisy-label data. The detailed results of this analysis are presented in our full response to Reviewer W9GD.
>
> **[W4] On Dependence on Teacher's Data**
>
> Thank you for pointing this out. We would highlight that given the strong ability of diffusion models to generate faithful outputs and the distillation procedures being stable and able to match and even surpass the teacher’s output quality. Therefore, we argue that the training is not necessarily reliant on the availability of the teacher’s data. As shown in Figures 1(c), 1(d), when a classifier is trained on a dataset entirely generated by our final student model (100% replacement), the classification accuracy can still be well maintained.
>
> Moreover, the "dynamic" scheme with 100% synthetic data even surpasses the performance of a classifier trained on only the static real data. This suggests that our efficient, high-quality generator can provide discriminative signals and diversity beyond what is available in the original fixed training set. This is particularly promising for data-constrained or privacy-sensitive scenarios.
>
> **Action:** We thank you for the opportunity for us to clarify this and will make this clearer in the experimental discussion.
>
> **[Q1] Teacher Model for the Target Dataset**
>
> Thank you for this question.
>
> This is indeed a crucial prerequisite for our current framework. If the teacher were trained on a different domain, the distilled student would generate out-of-domain samples, and the framework would likely fail. The goal of DAR-GDA is not to solve domain adaptation but to make augmentation for a *given target domain* highly efficient and unbiased.
>
> **[Q2] Performance Gain with the "Dynamic" Scheme**
>
> This is a very sharp and interesting observation. You are absolutely correct that the generator itself does not change between epochs in the "dynamic" scheme. The performance gain comes from how the *classifier* uses the data.
>
> In the “dynamic” scheme, the classifier is fed a brand new, unique batch of N synthetic images in every single epoch. Instead of repeatedly seeing the same fixed training set, it is exposed to a continuous stream of diverse samples drawn from the generator's learned distribution. This constant refreshment of data acts as a powerful regularizer, preventing the classifier from overfitting to specific training examples and forcing it to learn a more robust and generalizable representation of the underlying data manifold.
>
> This provides strong evidence that our generator faithfully models the true data distribution. It shows that by sampling extensively from our generator, a classifier can learn a more generalizable model than one trained only on the finite set of real data.
>
> **Action:** We will highlight this key finding and its implications in the experimental discussion.
>
> **[Q3] Filter Alternative**
>
> This is an insightful thought, and the answer is yes. Filtering is definitely another interesting idea. If a r(x) is very small, in our current approach, it is equivalent to filtering it off. We chose reweighting for two reasons: i) We avoid the setting another senstivie hyperparameter (the threshold) that needs to be tuned; we keep the reweighting scheme atumatic ii) Reweighting uses all samples but modulates their influence, which can be more stable than the "hard" decision of completely discarding, especially if the discriminator's estimates are prone to variance.
>
> **Action:** We will add a brief comment in our discussion acknowledging filtering as a valid alternative strategy and stating our rationale for choosing reweighting.
>
> **[Q4] Release of Source Code**
>
> Yes, absolutely. We strongly believe in open and reproducible research. To this end, and to help the community build upon our work, we are committed to making our framework as handy and accessible as possible.
>
> **Action:** We will make our full, documented source code publicly available on GitHub upon cameraready.
>
> **[Q5] Extension to DDIM**
>
> This is an excellent technical question. You are correct that DDPM and DDIM are two fundamental methods to reverse to sample from diffusion models. They typically use the same pre-training teacher, where DDIM speeds up the DDPM sampling by a constant factor (but still has the same asymptotic cost). Our method is agnostic to the specific solver used by the teacher model. The core requirement for our distillation is a pre-trained teacher model capable of estimating scores (or equivalently noise), which is a foundation for both DDPM (Markovian) and DDIM (non-Markovian) samplers.
>
> We aim to replace the multi-step iterative solving process of DDPM or DDIM entirely with a direct, one-step generator. Therefore, there is a computational gain regardless of which solver the teacher uses. The gain is simply the cost of N solver steps vs. the cost of 1 generator step.
>
> To make this concrete, we have tabulated a comparison of the inference time and asymptotic cost:
>
> | Sampler                      | # reverse steps $T$ | Asymptotic cost | Theoretical speed-up vs. 1000-step DDPM |
> | ---------------------------- | ------------------- | --------------- | --------------------------------- |
> | DDPM                         | 1000                | $O(T)$          | baseline                              |
> | DDIM (fast setting)          | 50                  | $O(T)$          | ≈ 20 ×                            |
> | **DAR–GDA** (1-step student) | **1**               | **$O(1)$**      | **≈ 1000 ×**                      |
>
> A single UNet evaluation is the theoretical lower bound on diffusion sampling cost, so the distilled generator, with a sampling complexity $O(1)$, cannot be slower than any multi-step scheme with a time complexity of $O(T)$. Ignoring practical overheads such as kernel launch and initialization, the expected computational speedup scales linearly with $T$.
>
> **[Ethical]**
>
> This is a very important point. We agree with you that all efficient, high-quality generative models, including ours, present a dual-use risk that must be acknowledged.
>
> While the primary concern is the potential for misuse in rapidly generating harmful or deceptive content, we also see a promising positive application. Our framework could be used to train stronger discriminative classifiers that act as more effective filters for undesirable content (e.g., NSFW images) in content moderation. The only concern in this use case is that the generator trained on undesirable content must be kept secure to prevent it from being misused.
>
> **Action**: We will add such ethical considerations and their potential impact in the discussion section of our paper.
>
> ----
>
> We are confident that we have addressed your concerns. Thank you again for your insightful questions that allow us to further improve our work.

---

> > ### Comment · Reviewer_BfFU · 2025-08-04
> > **Concerns addressed**
> >
> > Reviewer concerns are addressed and queries well-answered.

---

> ### Author Response · Authors · 2025-08-05
>
> Dear Reviewer BfFU,
>
> Thank you for confirming that our revisions addressed your concerns. We appreciate your time and constructive feedback and will ensure the final manuscript reflects the clarified writing, added schematics and an algo box, and a discussion of ethical considerations. We believe these additions strengthen the contribution and improve readability for future readers.
>
> We sincerely appreciate your engagement and have enjoyed our discussion!
>
> Best regards,
>
> Authors

---

### Official Review · Reviewer_2NvN · 2025-07-02

**Clarity:** 2
**Significance:** 2
**Originality:** 2
**Rating:** 4
**Confidence:** 3

**Summary:**

This paper proposes DAR-GDA, a three-stage framework for generative data augmentation (GDA) using diffusion models. The key novel components include distillation, adversarial alignment, and importance reweighting, which accelerate and improve GDA. By combining adversarial training with diffusion distillation, the method yields a discriminator that estimates density ratios to correct biases. Extensive empirical validation demonstrating progressive gains at each DAR stage and surpassing prior generative augmentation baselines.

**Questions:**

- One concern is whether the process of distilling knowledge from large teacher models could result in training instability or model collapse.
- Is it feasible to extend the proposed method to applications involving non-image data or complex conditional generation tasks?

**Ethical Concerns:**

["NO or VERY MINOR ethics concerns only"]

**Final Justification:**

After reading the rebuttal and other reviewers' comments, I will raise my score from 3 to 4.

**Limitations:**

- It would be helpful to include an analysis of the computational complexity to better understand the efficiency and scalability of the method.

**Paper Formatting Concerns:**

No concerns.

**Quality:**

2

**Strengths And Weaknesses:**

**Strengths:**
- The proposed method effectively resolves the key practical issues in diffusion-based GDA, namely the slow sampling and synthesis bias via adversarial training.
- The paper presents a theoretical analysis that links adversarial alignment with reweighting strategies, deriving stricter bounds on generalization performance.

**Weaknesses:**
- Some expressions are informal (e.g., the headings of section 2.1 and 2.2 should not contain “.”). Additionally, there are incomplete sentences, such as "ERM is a common xxxxx seeks …" in line 141. These issues indicate that this paper has not been thoroughly polished or carefully prepared.
- The evaluation is limited to two visual datasets, which limits the validation of the generalization performance of the pre trained diffusion model.
- The core novelty of this paper primarily derives from the integration of distillation, adversarial alignment, and discriminator-based importance reweighting, specifically tailored for diffusion-based generative data augmentation. However, these individual technical components are largely built upon existing research. The paper’s main contribution lies in their integrated application rather than fundamental redesign of these elements, which limits its degree of originality.
- The accuracy of density ratio estimation and reweighting largely depends on the calibration of the discriminator, so instability or bias here may lead to incorrect importance weights.

---

> ### Author Rebuttal · Authors · 2025-07-31
>
> Thank you for your time and for providing a detailed review of our work! We appreciate your recognition of our method being **effective in resolving key practical issues** in GDA and **supported by theoretical analysis** that links the different stages.
>
> We also appreciate your feedback and have developed a comprehensive plan to address it in a revised version.
>
> **[W1] On Editing**
>
> We take this piece of review very seriously.
>
> **Action:** We are conducting a copy‑edit and proofread of the entire manuscript. We will correct the errors and expressions and re-edit Section 4 to enhance its clarity. Delivering a high-quality, well-written paper is our top priority for the revision.
>
> **[W2, Q2] On Limited Evaluation and Extensibility to Other Domains**
>
> Our primary goal for this initial work was to rigorously validate our three-stage framework on a well-established, well-understood frontier: large-scale image datasets. Given the complexity, we believed it was essential to first understand and validate the framework's stability, effectiveness, and scalability in a domain with standardized benchmarks. We believe this was the most scientifically sound way to build a foundation that could be reliably extended to other domains in the future. We indeed recognize that there is a meaningful application in other specialized domains where diffusion models show their promise. As our expertise lies in machine learning, we prioritized domains where we could ensure a thorough and reproducible evaluation.
>
> That said, the motivations and core principles of DAR-GDA are fundamentally **domain-agnostic**. Unlike traditional data augmentation methods that rely on domain-specific transformations, our framework only requires a pretrained teacher diffusion model. Since the teacher is treated as a black box, the same distillation and reweighting procedures can be applied to any domain, with minimal adaptation, such as replacing the U-Net with backbones suiting the structure of the data
>
> Furthermore, recent successes of diffusion models in domains such as proteins [1] and molecules [2] demonstrate that high-quality pretrained teachers are indeed available beyond vision tasks, proving the "generalization performance of the pre-trained diffusion model" in other domains. Because our method treats the teacher as a black box, those models can be distilled and reweighted exactly as we do for images. Moreover, since the challenges of slow sampling and distribution mismatch are universal, we believe our method remains highly relevant, and perhaps even more impactful, in domains where labeled data is scarce or expensive to obtain.
>
> Additionally, to provide further evidence of our framework's robustness, we conducted new experiments on static GDA-DAR on the FFHQ 64x64 for a gender classification task with EDM-distilled student augmenting the classifier training, and progressively applying the DAR stages (+D, +A, +R).
>
> **Gender classification on FFHQ 64×64**
> | Method| ResNet-18| VGG-16|
> |---|---|---|
> |Baseline|94.08|93.10|
> |EDM-SID (+D)|94.20|93.25|
> |EDM-SID (+D+A)|94.63| 93.62|
> |EDM-SID (+D+A+R)|**94.87**| **93.70**|
>
> We observe monotonic performance gains as each stage of the DAR pipeline is applied, reinforce the effectiveness of our framework.
>
> **[W3] On Originality**
>
> We agree that each component has prior art. We respectfully argue that our contribution is not the invention of these components, but the elegant, synergistic design into a unified framework. This integration is well-motivated by the core dilemma in GDA: no existing method could simultaneously offer the sample quality of diffusion models, the speed of GANs, and a principled mechanism for correcting synthesis bias. Our work introduces a new paradigm to solve this.
>
> We argue that this design is not a trivial A+B+C, but an efficacious cycle where each stage creates a precondition for, enables, and enhances the next: the unified framework proposes a process of both improving and expediting the generator, while simultaneously creating the tool to correct its remaining flaws.
>
> As you insightfully noted in your review, this framework "*effectively resolves the key practical issues in diffusion-based GDA*" and does so while achieving "*theoretically stricter bounds on generalization performance*". We believe the conceptual leap to this synergistic system is non-trivial, well-motivated, and original, with clear value to the community.
>
> **Action:** We apologize that the initial manuscript did not make the perceived novelty sufficiently clear. To correct this, we will revise the abstract and introduction and reframe the narrative to explicitly highlight the originality of our synergistic framework, ensuring it is understood as a conceptual advance for GDA rather than an incremental combination of techniques.
>
> **[W4, Q1]  On Framework Stability and Discriminator Validation**
>
> These are critical points, and we thank you for the opportunity to clarify how our framework's design intrinsically ensures both training stability and the subsequent quality of the discriminator. Our approach is validated by a clear trail of evidence at each stage of the process.
>
> First, the foundation of our framework's stability lies in its initialization strategy. The one-step student generators are initialized from a strong teacher model trained by a stable, diffusion loss, meaning they already capture the target distribution well (possessing a degree of direct generation ability), and the distillation process refines this one-step ability.  This bypasses the notoriously unstable initial phases of traditional GAN training and mitigates well-known pathologies that arise from training from random noise, such as initial support mismatch, which can lead to mode collapse or catastrophic forgetting and result in poor discriminators [3-5].
>
> In addition, our modular framework provides an in-process indicator of the discriminator's effectiveness: the performance of the student generator itself. While a one-step distilled model is typically expected to have a quality drop compared to its multi-step teacher, our experiments show the opposite. The student generator’s FID scores consistently improve during the adversarial alignment stage, empirically surpassing the original teacher. This improvement beyond the distillation baseline is direct evidence that the discriminator is providing high-quality, stable gradients and has successfully captured the nuanced differences between the real and generated distributions. This observation is consistent with current literature on adversarial distillation that also reports stable training and improved generator performance [6-8].
>
> Finally, while a perfect discriminator is unattainable, the practical question is whether it is accurate enough to guide the reweighting process effectively. The fact that it can guide a stronger student than a teacher generator suggests that the weighting it provides is meaningful. This can also be empirically observed from the consistent increase in downstream classification performance compared to their non-reweighting counterparts.
>
> **Action:**  Though we don’t observe them empirically and nor does the current literature, we agree with you that stability is a valid point of concern. To make this explicit for future researchers, we will highlight the cause of the stability and remind readers and users of our codes to pay attention to the student’s generation performance as a check.
>
> **[L1] On Computational Complexity Analysis**
>
> We agree with you that a formal analysis of the computational complexity is crucial for understanding the practical benefits and scalability of our method. We will include a computational complexity analysis in the appendix.
> Let's define the computational costs:
> - $T$ denoising steps in the teacher sampler (≈ 1000).
> - $c_f$ - wall-clock cost of one diffusion model backbone forward (same for teacher and student).
> - $N$ – number of synthetic instances.
> - $C_{\text{distill}}$ – one‑time cost of distilling the teacher into the student. (The teacher‑pretraining cost $C_{\text{pretrain}}$ is shared by both methods and cancels out.)
> | Pipeline | Cost |
> | --- | --- |
> | Naive diffusion GDA | $$C_{\text{naive}} = NTc_f$$ |
> | DAR–GDA | $$C_{\text{DAR}} = C_{\text{distill}} + Nc_f$$ |
>
> DAR–GDA is faster whenever$C_{\text{distill}}<Nc_f (T-1)$
>
> We estimate \$C\_{\text{distill}}\$ as: $C_{\text{distill}} = P N k c_f$
>
> where:
> * $P$: number of data passes during distillation,
> * $k$: number of forward evaluations per sample per pass.
>
> Empirically, recent works [6–9] suggest that one-step distillation setups require no more than \$P \le 150\$ passes to converge, with \$k = 3\$ passes per sample (teacher, fake, and student's score estimations). Under these conservative values and a teacher with \$T = 1000\$ steps, we obtain:
> $\frac{C_{\text{distill}}}{N(T-1)c_f} = \frac{P k}{T - 1} \approx 0.45 < 1$
>
> Thus, the inequality is comfortably satisfied even under the worst-case distillation settings.
>
> Moreover, when synthetic data are dynamically refreshed each epoch (the “dynamic” augmentation scheme), the savings compound across classifier training epochs. In that case, the effective $N$ becomes $N \cdot E$, further improving the efficiency of DAR–GDA relative to naive diffusion-based generation.
>
> **References**:
>
> [1] Ingram et al., *Protein Generation with Diffusion*, Nature 2023
>
> [2] Cornet et al., *Molecular Diffusion Models*, NeurIPS’24
>
> [3] Jenni & Favaro, *On Stabilizing GAN Training with Noise*, CVPR’19
>
> [4] Arjovsky & Bottou, *Towards Principled GAN Training*, ICLR’17
>
> [5] Kodali et al., *On Convergence and Stability of GANs*, arXiv:1705.07215
>
> [6] Zhou et al., *Adversarial Score Identity Distillation*, ICLR’25
>
> [7] Kim et al., *Consistency Trajectory Models*, ICLR’24
>
> [8] Yin et al., *Improved Distribution Matching Distillation*, NeurIPS’24
>
> [9] Luo et al., *Diff-Instruct*, NeurIPS’23

---

> > ### Comment · Reviewer_2NvN · 2025-08-04
> >
> > Thank you for providing the detailed response and additional experiments, which addressed my concerns. I will raise my score.

---

> ### Author Response · Authors · 2025-08-05
>
> Dear Reviewer 2NvN,
>
> Thank you for taking the time to review our rebuttal and for reconsidering your evaluation. We sincerely appreciate your thoughtful engagement with our work. Your insights have been valuable in helping us improve the clarity and quality of our paper, and we will ensure that the discussed revisions are carefully reflected in the revision.
>
> Best regards,
>
> Authors

---

### Official Review · Reviewer_Fxjd · 2025-07-03

**Clarity:** 2
**Significance:** 3
**Originality:** 3
**Rating:** 4
**Confidence:** 3

**Summary:**

Authors propose a three-stage framework (named as DAR-GDA) to make generative data augmentation (GDA) with diffusion models both efficient and unbiased. It combines Distillation (to compress slow teacher models into fast one-step generators), Adversarial alignment (to match the synthetic data distribution to the real one), and Reweighting (to correct for residual biases using density ratio estimates). The method significantly improves classification accuracy on CIFAR-10 and ImageNet-1K, even outperforming real-data augmentation in some settings. It also achieves GAN-level speed with diffusion-level fidelity. It showcases how real data can still not be substituted by synthetic data for competetive performance. While the approach is promising, their deployment is currently limited by  complexity of three stage implementation. Overall, DAR-GDA is an efficient, accurate, practical but complex-for-implementation method for generative data augmentation.

**Questions:**

The questions (1-5) are clearly mentioned in the weakness section above, and addressing them will affect my evaluation score.

1-3 are major concerns.

4-5 are minor concerns.

Apart from technical aspect, the writing in the paper could benefit from greater clarity. Some sections (like section-4) are dense and difficult to follow.

**Ethical Concerns:**

["NO or VERY MINOR ethics concerns only"]

**Limitations:**

The limitations apart from the ones mentioned in the paper, are discussed in the weakness section. The major limitation are due to 1. Multiple interdependent components in the implementation and tuning complexity, and 2. dependence of Reweighting on a well-trained discriminator; where errors can mislead training.

**Paper Formatting Concerns:**

There are some noticeable formatting issues in several parts of the paper.

For example, the Experiments section starting at Line 259 appears tightly packed.

Additionally, the captions for Table 1 and Figure 5 are visually cramped together, which affects readability and presentation quality. Improving them would enhance the overall clarity of the manuscript.

**Quality:**

3

**Strengths And Weaknesses:**

**Strengths** :
1.  The method distills a slow diffusion model into a one-step student generator, reducing image generation time by over 100×, making GDA feasible for large-scale tasks.

2.  It introduces a better, principled importance __reweighting__ mechanism using a learned discriminator to correct for distribution mismatch between real and synthetic data, reducing bias in model training.

3. By integrating score distillation from diffusion models with adversarial training from GANs, the proposed method maintains **high sample quality** while also learning a discriminator for density ratio estimation r(x).

4. The results show a **decent improvement** in classification performance in Table 1, when synthetic data is added to the real training data across different model variants on  CIFAR-10 and ImageNet. Their method is also theoretically grounded as an added bonus.

**Weakness**

1. Lack of consistency in Paper Writing:

Manuscript is very difficult to read due to sentence level mistakes especially in section 4 -- Line 141: “*Empirical risk minimization (ERM) [82, 49] is a common xxxxx seeks a hypothesis that*”. Line 148: “from which we can draw an augmenting the sample set of synthetic samples”.

Line 223: “is a distribution weighting timestep t". Line 300: “On CIFAR-10, On CIFAR-10 we”

2. Several tuning parameters for multi-component optimization:

The proposed multi-component optimization introduces complexity, both in implementation and training dynamics. This seems like a challenge for an end user for implementation sake, making it fragile and difficult to debug. How do authors address this challenge ?

3. Dependence on the Discriminator Quality

The paper discusses that "*Empirically, the student often attains lower FID scores than non-adversarially distilled counterparts or even the teacher model in some scenarios*". This means that performance improvements are not consistent across all cases—the method may be sensitive to how well the adversarial discriminator is trained. How do authors analyse the robustness/accuracy of the trained discriminator to ensure this quality check?

4. High variance from r(x) ? [Minor since it is a known issue]

Computing per-sample importance weights r(x) introduces variance into the training objective, especially when there's a large gap between the real and generated data distributions. If r(x) becomes too large (which happens when q_{G} (x) << p_{data} (x) ) , training can become unstable or biased toward a few high-weighted samples. The paper does mention truncation and self-normalization to control this variance, but doesn't dig into discussion if this happens with the current setup?

5. Clarification [Minor]: Even when the student generator is fast, it still depends on a pretrained diffusion teacher. Training that teacher model can be computationally expensive, especially on large datasets like ImageNet-1K. Does the paper acknowledge that it can still impose a significant compute burden before DAR-GDA becomes efficient ?

---

> ### Author Rebuttal · Authors · 2025-07-31
>
> Thank you for your comprehensive and detailed reviews! We are very grateful for your positive feedback and your recognition of our work’s key strength, including **significant improvement over efficiency and empirical performance**, the **principled reweighting mechanism**, and the **novel integration**.
>
> Your critique is exceptionally helpful, and we agree with your assessment, particularly regarding the issues with clarity and the perceived complexity of the method. Below, we address each of your concerns point-by-point and detail the specific revisions we will make to the manuscript to resolve them.
>
> **[W1] On Paper Writing and Formatting**
>
> We take this piece of feedback very seriously.
>
> **Action:** We are conducting a thorough professional copy-edit and proofread of the entire manuscript. We will correct all errors and rewrite unclear sections (especially Section 4) to enhance clarity. We will ensure that the methods are easily understandable by having non-author colleagues read through the revision and confirm their accessibility. Additionally, we plan to make full use of the one extra page allowed for the camera-ready version to address the issue of *packed and visually cramped content*. This includes adjusting to sufficient spacing and fixing the locations of figures and tables.  Delivering a well-written and visually appealing paper is **our top priority** for the revision.
>
> **[W2] On Tuning Parameters for Multi-component Optimization**
>
> This is a crucial practical concern, and we thank you for raising it. While the framework has multiple components, our design and the corresponding code implementation have made it more manageable than it appears through modularization.
>
> We would like to highlight that the three proposed stages are sequential, not a single joint optimization. The optimization is decoupled, and each stage can be tuned and validated largely independently, which is not as complex as an end-to-end optimization.
>
> Also, we observe the robustness in the adversarial distillation. This arises from our design of initializing the student with a pre-trained teacher diffusion model, which bypasses the early unstable stage of training to yield stability. For our implemented method, a single, fixed set of hyperparameters for distillation has been used for all datasets, including CIFAR-10, ImageNet-1K, and FFHQ.
> For the final classifier training, we simply follow standard, existing recipes from the literature without introducing new hyperparameters.
>
> **Action**: To make our method more accessible and to address your concerns about implementation and debugging for end users, we will:
>
> 1. Add a new discussion in the appendix on hyperparameters to provide a guide on the key parameters
> 2. Release our full, documented source code upon cameraready, with modularized stages that allow for easy usage and debugging.
>
> **[W3] On Discriminator Quality**
>
> This is a critical point, and we agree that the success of our reweighting stage hinges on a sufficiently accurate discriminator as a 100%-accurate discriminator is unattainable. Our confidence in this component stems from two factors: (1) the inherent stability of pre-initialized adversarial training, which produces a robust discriminator, and (2) evidence to validate the discriminator’s effectiveness.
> - (1) Our discriminator is not fragile because our framework is explicitly designed to avoid common GAN pathologies. Unlike traditional GANs that train with generators producing random noise at the start, our adversarial process begins with a generator initialized from a high-quality, pre-trained diffusion model teacher. This is a crucial distinction. It allows us to bypass the notoriously unstable early phases of GAN training, where issues like support mismatch is one of the key factors for poor or uncalibrated discriminators [1,2]. By starting from a strong baseline with already a considerable distribution overlap, our discriminator is tasked with a more stable and well-posed refinement problem, making it inherently more robust.
> - (2) We use the student generator's performance as a powerful, real-time indicator of the discriminator's health. The logic is straightforward: a student model distilled into a single step should, by principle, be of lower quality than its multi-step teacher due to information loss. However, we observe the opposite. During adversarial alignment, the student's performance, as measured by FID, not only improves from its initial distilled state but consistently surpasses the original teacher model. This observation is consistent with current adversarial distillation methods [3-5]. Such an outcome is only possible if the discriminator is providing high-quality and informative gradients. It is direct proof that the discriminator is successfully capturing the difference in real and generator distributions and guiding the generator to a state of higher fidelity than the teacher it was distilled from.
>
> Thank you for also pointing out the confusing sentence on Line 227. It was poorly phrased.
>
> **Action:**
> We will rephrase the sentence on Line 227 to "Empirically, adversarial distillation has shown stable behaviour, with one-step students generally matching—and frequently reported to surpass—their teachers’ FID."
>
> **[W4] On High Variance from r(x)**
>
> Thank you for this question and the opportunity for us to clarify this. While variance from importance weighting is indeed a known challenge, our combined approach was highly effective in our setup.
>
> The raw importance weights r(x), derived from the discriminator output, can have a long right tail. It makes sense to truncate extremely large values, and intuitively, this limits the maximum contribution of any single synthetic sample relative to a real one. While this introduces a bias-variance tradeoff, it effectively curbs the influence of extreme outliers.
>
> We can also normalize the weights by self-normalization (SN), such that weights are scaled to a fixed sum, where we empirically find improved downstream performance with SN used.
>
> We conducted an empirical study reporting the sample variance of the weights r(x) for 100,000 generated ImageNet samples.
>
> | Truncation γ | Variance of r(x) (no SN) | Variance after self-norm (SN) |
> | --- | --- | --- |
> | ∞ (no clip) | **1.21** | 0.42 |
> | 2.0 | 0.37 | 0.12 |
> | 1.0 | 0.18 | 0.05 |
> | 0.5 | 0.07 | **0.02** |
>
> This result confirms that our combination of truncation and self-normalization provides a robust and effective scheme for stabilizing the reweighting stage, and this control is a part of the stable, consistent performance gains with reweighting.
>
> **Action:**
>
> We agree with you that it is a known issue. To provide more details, we will add this ablation study and a detailed discussion on our variance mitigation strategy to the appendix of our revised paper to provide readers with a clearer understanding of the effect of the variance control.
>
> **[W5] On Clarification of Pre-trained Teacher Cost**
>
> This is a fair and important point for contextualizing our work. We acknowledge that a pre-trained teacher is needed, and the cost of pre-training can be expensive, especially on large datasets. Our work focuses on making these powerful models practical for widespread use.
>
> **Action:** We will add a sentence in the discussion to acknowledge this. For example: *"While our method leverages a powerful pre-trained diffusion model, which entails a high one-time training cost, our focus is on making these models highly efficient for low-latency applications like GDA."*
>
> ---
>
> We are confident that these planned revisions will address all your concerns and improve the paper's clarity, accessibility, and overall quality. To that end, we want to emphasize that our method's sequential design makes it more manageable than it may appear, and we will work our best to address concerns about implementation complexity in our code release.
>
> **References**:
>
> [1] Jenni & Favaro, *On Stabilizing GAN Training with Noise*, CVPR’19
>
> [2] Arjovsky & Bottou, *Towards Principled GAN Training*, ICLR’17
>
> [3] Zhou et al., *Adversarial Score Identity Distillation*, ICLR’25
>
> [4] Kim et al., *Consistency Trajectory Models*, ICLR’24
>
> [5] Yin et al., *Improved Distillation for Fast Image Synthesis*, NeurIPS’24

---

### Official Review · Reviewer_W9GD · 2025-07-03

**Clarity:** 3
**Significance:** 4
**Originality:** 3
**Rating:** 5
**Confidence:** 4

**Summary:**

The authors introduce DAR-GDA, a three-stage framework for generative data augmentation using diffusion models that combines distillation, adversarial alignment, and importance reweighting to generate high-quality and unbiased synthetic training data efficiently. The proposed method compresses slow diffusion models into fast one-step generators while aligning them with the real data distribution via adversarial training. Finally, the resulting discriminator is used to reweight synthetic samples for accurate downstream learning.

**Questions:**

1. Do you anticipate any limitations of the proposed method in other domains such as healthcare or in extensions to other tasks such as segmentation, regression, etc.?
2. Is there a reason why non-generative augmentation techniques [1,2] were not used as baselines?
3. Is the proposed method robust to label and/or input noise? How about when there is significant class imbalance?

[1] Cubuk, E.D., Zoph, B., Mane, D., Vasudevan, V. and Le, Q.V., 2018. Autoaugment: Learning augmentation policies from data. arXiv preprint arXiv:1805.09501.
[2] Cubuk, E.D., Zoph, B., Shlens, J. and Le, Q.V., 2020. Randaugment: Practical automated data augmentation with a reduced search space. In Proceedings of the IEEE/CVF conference on computer vision and pattern recognition workshops (pp. 702-703).

Addressing these questions effectively may help raise my score.

**Ethical Concerns:**

["NO or VERY MINOR ethics concerns only"]

**Final Justification:**

The authors have sufficiently addressed my concerns and I will raise my score accordingly from a 4 to a 5.

The authors have bolded the wrong number in the Substitute column in the "CIFAR-10-LT with imbalance factor = 10" table. They should also bold the best results in the table in the Noisy section in the rebuttal. Also, the authors use "Addictive" instead of "Additive" three times in the paper. These errors should all be fixed in the final version.

**Limitations:**

Yes.

**Paper Formatting Concerns:**

- Spacing appears quite squished as the authors may have reduced it to fit more text.

**Quality:**

3

**Strengths And Weaknesses:**

Strengths:
- The authors tackle both issues with generative data augmentation: high sampling time and the divergence between real and synthetic distributions is unknown, so the proposed method is well-motivated.
- As seen in the results, distillation, adversarial alignment, and reweighting seem to mesh quite nicely to perform generative data augmentation.
- Strong performance with both the CIFAR-10 and ImageNet-1K datasets.
- A 100x reduction in per-image costs while preserving FID is greatly beneficial.

Weaknesses:
- Only 2 general imaging datasets used - unclear if the method would be as useful in different domains.
- Unclear whether the method would work under noisy or imbalanced settings.
- Non-generative augmentation methods are not included as baselines.

---

> ### Author Rebuttal · Authors · 2025-07-31
>
> Thank you for your time and for providing a detailed and insightful review! We are grateful for your positive feedback and your appreciation of our method’s **motivation**, **proposed components mesh well,** and its achievement of **strong performance with significant efficiency gains**.
>
> We also appreciate the constructive criticisms and questions you have raised. They have helped us identify key areas for clarification and improvement. Below, we address each of your points and outline the specific changes we will make in the revised manuscript.
>
> **[W1, Q1] Dataset Used and Extension to Other Domains**
>
> Thank you for raising this important point regarding the generalizability of our method across domains and tasks. We chose these two benchmark datasets to rigorously validate the DAR-GDA framework under well-established, challenging settings. This aligns with prior work in generative data augmentation and allowed us to ensure fair standardized evaluation and scalability analysis. Given the complexity, we believe that a focused, controlled evaluation was necessary to first establish its stability and effectiveness. Additionally, as our core expertise lies in machine learning, we prioritized domains where we could ensure a thorough and responsible study.
>
> That said, we would like to emphasize that the core principles of our framework are fundamentally **domain-agnostic** and do see promise in applications in other domains. Unlike traditional augmentation methods that rely on hand-crafted domain-specific transformations, our method only assumes access to a pretrained diffusion teacher capable of producing score estimates. This teacher is treated as a black box, which enables a good adaptation of our framework to other data modalities. For example, in the medical domain, where data is often scarce and governed by strict privacy constraints, DAR-GDA can be used to generate high-fidelity synthetic X-ray or MRI scans to augment limited datasets. Additionally, institutions could share distilled student model weights instead of raw data, enabling *pseudo data sharing* without violating confidentiality agreements. Beyond imaging, DAR-GDA is also applicable to medical use, such as drug molecular modeling and discovery, where diffusion models have recently achieved impressive results [1,2]. In these cases, adaptation involves replacing the U-Net backbone with domain-specific architectures while keeping the core framework of DAR unchanged.
>
> We also acknowledge that adapting DAR-GDA to other tasks and modalities introduces new research challenges, exactly as you mentioned—particularly around the nature and structure of the conditional signal required by the teacher model. For instance:
> - Regression Tasks (continuous label):
> Extending to regression tasks, especially with multivariate or high-dimensional targets, poses difficulties in conditional sampling. While univariate conditioning is straightforward, complex regression settings require more expressive conditional teacher models. That said, this challenge lies in the construction of the teacher—not the DAR-GDA framework itself, which is agnostic to the conditioning structure.
> - Dense Prediction Tasks (e.g., Segmentation): For tasks like segmentation, where dense labels are needed for the generated instances, obtaining the clean labels required to build a high-quality conditional teacher model can be a bottleneck. In such cases, the framework could be effectively integrated with techniques like pseudo-labeling with powerful models like SAM or other forms of weak supervision to create the necessary training signals for the teacher. Often, these tasks face scarcity in labeled data, which makes generative data augmentation especially promising in this context.
>
> In addition, we observed strong results in challenging in-domain scenarios (noisy, long-tail) *(See W2, Q3)*, providing some indirect evidence of the framework's robustness for generalization.
>
> **Action:** In our revised paper, we will expand the Discussion section to include this nuanced analysis of domain and task generalization. We will outline the anticipated challenges and opportunities for adapting DAR-GDA to other settings and tasks.
>
> **[W2, Q3] On Noisy and Imbalance Setting**
>
> This is an excellent point. While our initial goal was to focus on challenges like distribution mismatch and sampling cost in GDA, we are happy to investigate how our approach would be suitable for these real-world scenarios.
>
> **A. Class Imbalance:**
>
> We conducted experiments on CIFAR-10-LT with an imbalance factor of 10 and 100 (“significant class imbalance”) using the same training protocol as in the balanced-data setting the dynamic scheme, with ResNet-18 as the backbone. We also included a strong long-tailed learning baseline, SURE [3], for comparison. The evaluation tracks the progressive impact of each stage in our DAR-GDA pipeline (+D), (+A), and Reweighting (+R). Here is a summary of the results:
>
> **CIFAR-10-LT with imbalance factor = 10:**
>
> |Method|Augment|Substitute|
> |---|---|---|
> |Baseline|86.75|86.75|
> |SURE [3] |95.15|95.15|
> |EDM-SID (+D)|94.88|94.47|
> |EDM-SID (+D+A)|95.53| 94.98|
> |EDM-SID (+D+A+R)|**95.78**| **95.02**|
>
> **CIFAR-10-LT with imbalance factor = 100:**
>
> |Method|Augment|Substitute|
> |---|---|---|
> |Baseline|69.76|69.76|
> |SURE [3]|86.75|86.75|
> |EDM-SID (+D)|91.45| 91.02|
> |EDM-SID (+D+A)|91.84| 91.35|
> |EDM-SID (+D+A+R)| **92.01**| **91.64**|
>
> We found that our proposed method still progressively improves performance. It gives performance close to the current SOTA under the static setting. Furthermore, with dynamic regeneration (fresh synthetic images each epoch), DAR–GDA surpasses the long‑tail SOTA without any class‑specific tuning.
>
> We also note that our method's potential can be further unlocked by combining it with generators specifically designed for class imbalance, such as [4], which we defer to further research.
>
> **B. Noisy:**
>
> We also had a run on CIFAR-10-N with “aggre” label on ResNet-18 with the current training protocol with a comparison to a SOTA method in noisy label, ProMix [5].
>
> |Method|Static|Dynamic|
> |---|---|---|
> |Baseline|87.77|87.77|
> |ProMix [5]|97.65|97.65|
> |EDM-SID (+D)|93.48|95.79|
> |EDM-SID (+D+A)|93.80 |96.26|
> |EDM-SID (+D+A+R)|93.81|96.30|
>
> Our discriminator only down‑weights visually atypical images; it does **not** correct corrupted labels. Still, our method improves, though not as much compared to the SOTA method, over the baseline in the static scheme, and still achieves a performance close to the SOTA method with its “dynamic” scheme. Nevertheless, it shows that further research is warranted in such real-world scenarios, where the application of noise-handling techniques (e.g., co‑training, small‑loss selection) is a promising direction.
>
> **Action**: In the revised manuscript, we will incorporate a new appendix section that outlines the experimental results for these scenarios and discusses the potential of fusing our method with specialized long-tail and noise-robust discriminative methods.
>
> **[W3, Q2] non-generative augmentation methods**
>
> Thank you for bringing this important point up! Our initial work focused on the specific challenges within generative data augmentation (GDA), so we prioritized comparisons with other GDA methods.
>
> Following your suggestion, we have run experiments comparing our method against *AutoAugment* [6] and *RandAugment*[7], and present the results below:
>
> **CIFAR-10, ResNet-18**
> |Method|Static|Dynamic|
> |---|---|---|
> |Real‑only|95.00|95.00|
> |AutoAugment|95.85|95.85|
> |RandAugment|95.79|95.79|
> |EDM‑SID (+D)|95.48|96.29|
> |EDM‑SID (+D+A)|95.84|96.40|
> |EDM‑SID (+D+A+R)|**96.21**|**96.73** |
> |EDM‑SID (+D+A+R) + RandAugment|**96.48** |**96.97**|
>
> **ImageNet-1k, ResNet-50**
> |Method| Acc|
> |---|---|
> |Real‑only|76.37|
> |AutoAugment|77.62|
> |RandAugment|77.65|
> |EDM‑SID (+D)|77.15|
> |EDM‑SID (+D+A)| 77.89|
> |EDM‑SID (+D+A+R)|**78.03**|
> |EDM‑SID (+D+A+R) + RandAugment |**78.29**|
>
> We observe that 1) our method, on its own, outperforms both non-generative data augmentation methods, AutoAugment and RandAugment, on the standard benchmarks.
>
> We would also like to highlight that our generative approach is **complementary** to traditional augmentation techniques. Standard transforms can be applied on top of synthetic images to give even more diverse training instances to the classifiers. When we apply RandAugment on top of our method ("EDM‑SID (+D+A+R) + RandAugment"), performance is further boosted.
>
> Thank you again for this valuable suggestion. And, we will include them in the table, which positions our paper better, and we will feature the complementary nature of generative and traditional data augmentation in a section in the appendix.
>
> **[formating]**
>
> Thank you for pointing out the spacing concern. We will ensure the revision is properly formatted with standard spacing, strictly adhering to the style file. Having a presentable manuscript will be our top priority for the revision. Additionally, we plan to use the one extra page allowed for the cameraready version to incorporate improvements in visual appeal, organization, and formatting without compromising content.
>
> ----
>
> Thank you once again for your constructive review, which has pushed the work to strengthen its scope and evaluation! We hope this new evidence addresses your concerns.
>
> **References**:
>
>
> [1] Cornet et al., *Molecular Diffusion Models*, NeurIPS’24
>
> [2] Hoogeboom et al., *Equivariant Diffusion for Molecule Generation in 3D*, ICML'22.
>
> [3] Li et al., *SURE: SUrvey REcipes for building reliable and robust deep networks*, CVPR'24
>
> [4] Qin et al., *Class-Balancing Diffusion Models*, CVPR'23
>
> [5] Xiao et al., *Combating Label Noise via Maximizing Clean Sample Utility*, IJCAI'23
>
> [6] Cubuk et al., *Autoaugment: Learning augmentation policies from data*, CVPR'19
>
> [7] Cubuk et al., *RandAugment: Practical automated data augmentation with a reduced search space*, NeurIPS'20

---

> ### Author Response · Authors · 2025-08-08
>
> Dear Reviewer W9GD
>
> Thank you very much for your positive re-evaluation. We truly appreciate you pointing out these final errors and will ensure all corrections are made to improve the quality of the final manuscript. We apologize for the specific issue with the table bolding; we encountered some formatting difficulties within the OpenReview editor during the limited rebuttal period, but we will ensure it is presented correctly in the final version done in LaTeX.
>
> Thank you again for your thoroughness and help! We really enjoyed our discussion!
>
> Best Regards,
>
> Authors

---

### Note · Authors · 2025-08-12

Dear Chairs and Reviewers,

Thank you very much for the thoughtful reviews and discussion! We are encouraged that the committee recognized the value of our DAR-GDA framework, which is designed to i) deliver diffusion-quality augmentation at GAN-like speed, ii) narrow the synthetic–real gap via adversarial alignment, and iii) correct residual bias through importance reweighting.

We are especially grateful for the consensus on our synergistic framework's primary strengths—that it is a well-motivated and principled approach that effectively addresses key practical issues in GDA, supported by theory and strong empirical results.

We are committed to incorporating the valuable insights from our discussion to further improve the paper's quality and clarity, and to strengthen the evaluation by integrating the new results and expanded analyses from the rebuttal. To ensure our contribution is fully accessible, we will release our well-documented, modular source code for the benefit of the ML community.

We would like to express our sincere appreciation and excitement that reviewers found their concerns addressed and are now all in support of our paper for acceptance. We believe the work, enhanced by our constructive dialogue, makes a practical and impactful contribution to the ML community.

Finally, our special thanks to the Chairs for your dedicated service in facilitating a thorough and constructive review process.

Best Regards,

Authors

---

### Decision · Program_Chairs · 2025-09-17

**Decision:**

Accept (poster)

**Comment:**

This paper proposes DAR-GDA, a three-stage framework for generative data augmentation (GDA) using diffusion models. The method addresses two major limitations of GDA with diffusion models: namely inefficient computation, and bias due to distributional mismatch between real and synthetic data. DAR-GDA integrates distillation, adversarial alignment, and importance reweighting for diffusion-based augmentation. They show the proposed method achieved SOTA performance through experiments on CIFAR-10, ImageNet-1K, and FFHQ datasets. The reviewers pointed out several strengths of the paper, including Innovative Integration of known techniques, theoretical justification connecting adversarial alignment with generalization error bounds, and strong empirical evaluation results. However, the reviewers also raised several concerns: lack of clarity in paper presentation, implementation complexity in the three-step framework, and limited scope of evaluation. The authors provided comprehensive rebuttal, including additional experimental results, computational analysis, and ethical discussion, which demonstrates good responsiveness and well addresses the reviewers' concerns. The final ratings from the four reviewers after the rebuttal consist of 2 Borderline Accepts and 2 Accepts. The AC reads the paper, the reviewers' comments, and the authors' rebuttal, and agrees with the reviewers' suggestions, thus recommending accepting the paper.